# STEAD: Robust Provably Secure Linguistic Steganography with Diffusion Language Model

**Yuang Qi**[†], **Na Zhao**[†], **Qiyi Yao**, **Benlong Wu**,
**Weiming Zhang**, **Nenghai Yu**, **Kejiang Chen**[*]

University of Science and Technology of China
Anhui Province Key Laboratory of Digital Security

{qiyuang@mail., znzhaona@mail., chenkj@}ustc.edu.cn

Codes: https://github.com/7-yaya/STEAD

## Abstract

Recent provably secure linguistic steganography (PSLS) methods rely on mainstream autoregressive language models (ARMs) to address historically challenging tasks, that is, to disguise covert communication as "innocuous" natural language communication. However, due to the characteristic of sequential generation of ARMs, the stegotext generated by ARM-based PSLS methods will produce serious error propagation once it changes, making existing methods unavailable under an active tampering attack. To address this, we propose a robust provably secure linguistic steganography with diffusion language models (DLMs). Unlike ARMs, DLMs can generate text in partial parallel manner, allowing us to find robust positions for steganographic embedding that can be combined with error-correcting codes. Furthermore, we introduce an error correction strategies, including pseudo-random error correction and neighborhood search correction, during steganographic extraction. Theoretical proof and experimental results demonstrate that our method is secure and robust. It can resist token ambiguity in stegotext segmentation and, to some extent, withstand token-level attacks of insertion, deletion, and substitution.

## 1 Introduction

In the increasingly digital world, the relevance of linguistic steganography has grown significantly. It offers a discreet method for covert communication across various applications, including intelligence operations, secure corporate communications, and privacy preservation [1]. Furthermore, in regions with strict censorship policies, linguistic steganography can serve as a means to bypass content restrictions, enabling individuals to share and access information discreetly. Traditional linguistic steganography methods typically embed secret messages directly into existing text by modification [2, 3] or by training a specialized text generation model to produce steganographic text [4, 5]. However, these methods' security cannot be guaranteed, and they remain susceptible to detection by sophisticated steganalysis techniques [6, 7, 8, 9]. Alternatively, provably secure steganography (PSS) methods [10, 11, 12] theoretically guarantee the indistinguishability of the stego (the carrier containing hidden data) and the cover (the original unaltered carrier).

In recent years, the progress of generative artificial intelligence has underscored the potential of autoregressive models (ARMs) in the field of provably secure linguistic steganography [13, 14, 15, 16, 17]. ARMs [18, 19, 20] generate text sequentially by predicting the next token based on the preceding context. They have played a particularly significant role in advancing provably secure steganography due to the precise sampling they provide, a capability that is unattainable in natural language environments [21].

---

[*]Corresponding author; [†]: equal contribution

39th Conference on Neural Information Processing Systems (NeurIPS 2025).

However, despite the high quality of text generated by ARMs, their sequential generation characteristic makes previous provably secure steganography methods highly vulnerable [22, 23, 24, 25, 26]. In ARMs, the probability of each token being generated is conditional on all previously generated tokens, meaning that once a stego token is altered, it not only interferes with the message of the current token but also impacts all messages of subsequent tokens due to the change in the conditional distribution. In a strictly regulated environment, adversaries may tamper with the text transmitted over public channels, causing significant disruption to message extraction. Additionally, the ambiguity in the segmentation of stegotext further increases the likelihood of errors during message extraction [27, 28, 29, 30].

Recently, the rapid development of discrete diffusion language models (DLMs) [31, 32, 33, 34, 35, 36] has presented an opportunity to address the vulnerabilities in provably secure linguistic steganography. DLMs have gained considerable attention since their introduction to the field of text generation, emerging as a promising alternative for sequential generation. Unlike ARMs, which generate tokens sequentially, DLMs begin from a completely noisy state and dynamically refine the entire sequence in a parallel manner. This different generation paradigm can, to some extent, mitigate the error propagation caused by sequential generation, thereby enhancing provably secure linguistic steganography and addressing its lack of robustness. Furthermore, the growing popularity of diffusion language models also provides an ideal camouflage environment for steganography.

However, directly bringing the DLM into PSS will not solve the problem of error propagation. Specifically, the advantages brought by DLM's parallel sampling can only be realized within a single denoising step. The model distribution for the next step still depends on the previously sampled tokens. Directly deploying PSS to DLM does not bring the expected robustness, and, due to the strong binding between the distributions and the token positions, it will be more fragile than the PSS method based on ARM when faced with possible position offsets (such as additions, deletions, or token ambiguity). Therefore, correcting potential errors within each step is crucial for achieving robust steganography based on DLM.

In this paper, we propose a provably secure and robust linguistic **STEgA**nography method using a **D**iffusion language model, namely **STEAD** (serving as an acronym for the technique while metaphorically highlighting its "steady" nature). To achieve error-correctable message embedding within a single-step denoising process, we introduce a message-driven pseudo-random number sampling algorithm with a fixed embedding capacity, based on the trade-off between robustness and embedding capacity. The message is embedded only at denoising positions which satisfies the encoding conditions of repetition error-correcting coding. We refer as a robust position embedding with repetitive error correction coding. During extraction, in addition to decoding ECC, we also introduce a neighborhood search strategy to address position offsets caused by insertions and deletions.

**Contributions:** (1) we propose a provably secure and robust linguistic steganography framework based on discrete diffusion language models; (2) we design robust position embedding with repetitive error correction coding and neighborhood search strategy to enhance the robustness of steganography; and (3) theoretical proof and experimental results demonstrate the superiority of STEAD in terms of security and robustness.

## 2 Background

### 2.1 Preliminaries

**Definition 2.1** (Stegosystem). *A symmetric key steganographic system (stegosystem) is a triple $\mathcal{S} = (\mathcal{K}, \mathcal{E}, \mathcal{D})$, where (1) $\mathcal{K}(\lambda) \rightarrow \mathbf{k}$ is a probabilistic key generation algorithm that takes the security parameter $\lambda$ and outputs a symmetric key $\mathbf{k}$; (2) a probabilistic encoding algorithm $\mathcal{E}(\mathbf{k}, \mathbf{h}, \mathbf{m}) \rightarrow \mathbf{st}$ takes a key $\mathbf{k}$, a channel history $\mathbf{h}$ and a message $\mathbf{m}$ as inputs, and outputs a stegotext $\mathbf{st}$; and (3) a deterministic decoding algorithm $\mathcal{D}(\mathbf{k}, \mathbf{h}, \mathbf{st}) \rightarrow \mathbf{m}$ takes a key $\mathbf{k}$, a channel history $\mathbf{h}$, and a stegotext $\mathbf{st}$, and outputs a decoded message $\mathbf{m}$.*

**Definition 2.2** (Computational Security). *A stegosystem $\mathcal{S} = (\mathcal{K}, \mathcal{E}, \mathcal{D})$ is computationally secure under chosing hidden-text attack if, for any probabilistic polynomial-time adversary $\mathcal{A}$, the advantage of $\mathcal{A}$ in distinguishing between the output stegotext of the steganographic encoding algorithm $\mathcal{E}$ and the output of random sampler $\mathcal{O}$ is negligible. Formally, if*

$$\left| \Pr\left[ \mathcal{A}^{\mathcal{E}(\mathbf{k},\mathbf{h},\mathbf{m})}\left(\mathbf{st}, 1^{\lambda}\right) = 1 \right] - \Pr\left[ \mathcal{A}^{\mathcal{O}(\mathbf{h},\cdot)}\left(1^{\lambda}\right) = 1 \right] \right| < negl(\lambda), \tag{1}$$

*for all sufficiently large $\lambda \in \mathbb{N}$, where $negl(\lambda)$ is a negligible function that correlates to the security parameter $\lambda$, the stegosystem is computationally secure.*

**Definition 2.3** (Correctness). *A stegosystem $\mathcal{S} = (\mathcal{K}, \mathcal{E}, \mathcal{D})$ is correct if, for any message $\mathbf{m}$ and any history $\mathbf{h}$, there exists a negligible function such that $\mathcal{E}$ and $\mathcal{D}$ satisfy the relationship:*

$$\Pr\left[\mathcal{D}(\mathbf{k}, \mathcal{E}(\mathbf{k}, \mathbf{m}, \mathbf{h}), \mathbf{h}) \neq \mathbf{m}\right] < \delta, \tag{2}$$

*where $\delta$ is a correctness parameter, which is typically very close to 0.*

Under the threat model of limited tampering capability, robustness ensures that the secret message can be successfully extracted even if the stegotext is under attack by adversaries.

**Definition 2.4** ($\delta$-Robustness against $\mathcal{F}_{\alpha,\beta,\gamma}$-Tampering). *A stegosystem $\mathcal{S} = (\mathcal{K}, \mathcal{E}, \mathcal{D})$ is said to be $\delta$-robust if, for all probabilistic polynomial-time adversaries $\mathcal{A}$ who have the $\mathcal{F}_{\alpha,\beta,\gamma}$-bounded tampering ability, for any $f_t \in \mathcal{F}_{\alpha,\beta,\gamma}$, there exists a negligible function such that*

$$\Pr\left[\mathcal{D}(\mathbf{k}, f_t(\mathcal{E}(\mathbf{k}, \mathbf{m}, \mathbf{h})), \mathbf{h}) \neq \mathbf{m}\right] < \delta. \tag{3}$$

**Definition 2.5** (Pseudo-Random Number Generator). *A pseudo-random number generator (PRNG) is a deterministic polynomial-time algorithm such that takes a $\lambda$-bit seed $\mathbf{k} \in \{0,1\}^\lambda$ as input, and ouputs a $m(\lambda)$-bit string, where $m(\lambda) > \lambda$. There exists a negligible function $negl(\lambda)$ for any probabilistic polynomial time distinguisher $\mathcal{A}$, such that*

$$\left| \Pr_{\mathbf{k} \leftarrow \{0,1\}^\lambda}\left[\mathcal{A}\left(PRNG(\mathbf{k})\right) = 1\right] - \Pr_{\mathbf{u} \leftarrow \{0,1\}^{m(\lambda)}}\left[\mathcal{A}\left(\mathbf{u}\right) = 1\right] \right| < negl(\lambda), \tag{4}$$

*where the seed $\mathbf{k}$ is randomly selected from $\{0,1\}^\lambda$, and $\mathbf{u}$ is uniformly chosen from $\{0,1\}^{m(\lambda)}$.*

## 2.2 Related Works

**Masked diffusion language models.** Unlike autoregressive models, diffusion language models support parallel generation. They mask the original tokens with special symbols $\mathbf{M}$ with a certain probability during the forward diffusion process, and learn the denoising distribution during the reverse process, thereby generate complete text. The original sequence of tokens $\mathbf{x}_0 = [\mathbf{x}_0^1, \ldots, \mathbf{x}_0^L]$ of length $L$. Define a forward Markov process at time steps $s$ to $t$,

$$q(\mathbf{x}_t|\mathbf{x}_s) = \prod_{i=1}^{L}(\beta_t \mathbb{1}\{\mathbf{x}_t^i = \mathbf{M}\} + (1 - \beta_t)\mathbb{1}\{\mathbf{x}_t^i = \mathbf{x}_s^i\}), \tag{5}$$

where each token at position $i \in \{1, \ldots, L\}$ is replaced with the mask symbol $\mathbf{M}$ with probability $\beta_t$, $\mathbb{1}(\cdot)$ is the indicator function. The reverse process is given by a parameterized model $p_\theta(\mathbf{x}_s|\mathbf{x}_t) = \prod_{i=1}^{L} p_\theta(\mathbf{x}_s^i|\mathbf{x}_t, t)$. In practice, $p_\theta(\mathbf{x}_s|\mathbf{x}_t)$ is learned through a masked language model (such as BERT or a Transformer encoder), and a categorical distribution is output for each masked position.

**Provably secure steganography with autoregressive models.** Recently, researchers have proposed several provably secure linguistic steganography methods based on ARMs. These methods are dedicated to designing message embedding algorithms that are indistinguishable from the normal generation process, i.e., random sampling. Kaptchuk *et al.* [15] introduced the Meteor method based on interval reversibility using a random sampling process akin to arithmetic encoding. Ding *et al.* [16] utilized the concept of "sampled distribution" to express information and presented the Discop method based on distribution copies. This method defines a probability distribution, from which multiple distribution copies are created, and then uses the index values of distribution copies to express messages. Wang *et al.* [17] proposed SparSamp, achieving unambiguous message embedding and extraction with a high embedding rate and an added complexity of only $O(1)$. The above methods can achieve provably secure linguistic steganography based on ARMs. None of them alter the probability distribution of words to be generated by the model during the process of embedding secret messages. But at the same time, none of them are robust to any form of stego tampering.

**Comparision to Works Done Concurrently** The works conducted concurrently with this paper have also noted the lack of robustness in linguistic steganography. Perry et al. [22] have fully discussed the robust steganography of LLMs but have ignored the concern for steganographic security. Wu et al. [25], who also propose a DLM-based robust linguistic steganography, adopt a coverless paradigm, leading to inconsistent research scope with ours. Bai et al. [23] proposed a provably secure steganographic method focused on asymmetric resource scenarios (PSARS), which achieves certain robustness while achieving provable security.

# 3 Threat Model

In this section, we define the adversary's goals and capabilities in robust steganographic scenarios.

## 3.1 Adversary Motivation

We consider a probabilistic polynomial-time (PPT) adversary operating in a highly regulated environment, responsible for detecting and disrupting covert communication behaviors that may be present in text publications. This involves two key capabilities: first, the ability to detect steganographic text, corresponding to the well-established field of steganalysis, which is concerned with the security of steganography; second, the ability to prevent the steganographic recipient from extracting the secret message, which relates to the robustness of steganography. Since steganographic texts are often transmitted over public channels, this can be achieved through a Man-in-the-middle attack. In this attack, the adversary intercepts the text that the steganographic sender attempts to publish, alters it, and then republishes it, with the receiver only able to access the tampered message. In real-world Internet scenarios, a potential Man-in-the-middle adversary could be a social media operator or an email service provider that is subject to governmental policy requirements.

However, the adversary's ability to modify the text provided by the sender is not without limitations. Excessive regulation or censorship may provoke strong public opposition, and overly stringent policies are often difficult to enforce at a low cost. Internet service providers, driven by business interests, also need to respect users and cannot engage in unlimited alteration of uploaded content. Therefore, the adversary is not aiming to fundamentally alter the content of the text, which creates a trade-off in its ability to interfere with the extraction process.

## 3.2 Adversary Capabilities

**Detection ability.**    In a generative linguistic steganography scenario, the ability of a PPT adversary to detect can be defined as the advantage in distinguishing between the output of a steganographic embedding algorithm (stegotext) and the original output of the model (covertext).

**Tampering ability.**    The ability of an adversary to tamper with stegotexts can be defined as a tampering function $f_t$ which ensures that for any stegotext $\mathbf{x}$ as input, the tampered text $f_t(\mathbf{x})$ satisfies that the extent of substitution, insertion, and deletion of words does not exceed $\alpha$, $\beta$, $\gamma$, respectively. Formally, let $\mathcal{V}$ be the vocabulary and let $\mathbf{x} = (x^1, x^2, \ldots, x^L) \in \mathcal{V}^L$ be the input text of length $L$. Define the tampering function $f_t : \mathcal{V}^L \to \mathcal{V}^*$ and denote its output by $\mathbf{y} = f_t(\mathbf{x}) \in \mathcal{V}^*$, where $\mathcal{V}^*$ is with an uncertain length. Let $N_{\text{sub}}, N_{\text{ins}}, N_{\text{del}}$ be the number of substitutions, insertions, and deletions, respectively. Then for given parameters $0 \leq \alpha, \beta, \gamma \leq 1$, $f_t$ is required to satisfy $N_{\text{sub}}(\mathbf{x}, \mathbf{y}) \leq \alpha L$, $N_{\text{ins}}(\mathbf{x}, \mathbf{y}) \leq \beta L$, $N_{\text{del}}(\mathbf{x}, \mathbf{y}) \leq \gamma L$. The tampering function is required to satisfied $f_t \in \mathcal{F}_{\alpha,\beta,\gamma}$, where $\mathcal{F}_{\alpha,\beta,\gamma}$ is a feasible set of functions,

$$\mathcal{F}_{\alpha,\beta,\gamma}(\mathbf{x}) = \left\{ \mathbf{y} \in \mathcal{V}^* \,\middle|\, N_{\text{sub}}(\mathbf{x}, \mathbf{y}) \leq \alpha L, \ N_{\text{ins}}(\mathbf{x}, \mathbf{y}) \leq \beta L, \ N_{\text{del}}(\mathbf{x}, \mathbf{y}) \leq \gamma L \right\}. \tag{6}$$

# 4 STEAD Methodology

In this section, we will first define the concept of a secure and robust steganographic system and introduce how to utilize the different characteristics of diffusion language models and autoregressive models to find the robust positions for steganographic embedding. Then, we will introduce a provably secure linguistic steganographic method, STEAD, that can achieve robustness against insertion, deletion, substitution, as well as token ambiguity. In this paper, we focus on symmetric key steganographic systems: the sender and receiver share the same settings, including an exact same diffusion language model, an initialized prompt, a pre-negotiated key (seed) of pseudo-random number generator, and a set of sampling parameters.

Figure 1 shows the embedding and extraction process of steganography using STEAD. Specifically, in the embedding stage, we proposed "robust position embedding with error correction coding" for DLM. We only embed messages at positions that are simultaneously and independently denoised to avoid cumulative errors. In each step, we select a fixed embedding capacity based on the minimum entropy of the distribution and use repetitive codes as error correction codes (ECC) to correct errors while minimizing interference between different embedding positions. In the extraction process, in

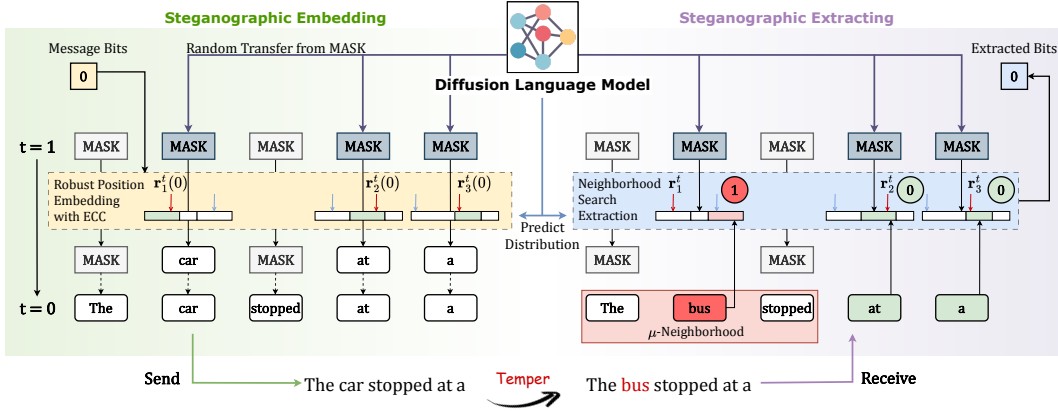

Figure 1: An overview of the proposed STEAD stegosystem.

addition to decoding ECC, we also introduce a neighborhood search strategy to address the position offset caused by insertion and deletion.

## 4.1 Diffusion Sampling Process

To achieve robustness in linguistic steganography, we innovatively refer to the multiple positions of tokens that can be synchronously independently sampled in one single timestep of the DLM as a batch of **robust positions** suitable for steganography, which is significantly different from the generation process of the ARM. To generate a token sequence of length $L$, the reverse diffusion process starts with the sequence $\mathbf{x}_{t=1}^{1:L}$ where $\mathbf{x}_{t=1}^i$ is fully masked by a special token $\mathbf{M}$ for all $i \in \{1, \ldots, L\}$. For timestep $t$ to $s$ with $0 \leq s < t \leq 1$, the conditional distribution for the reverse process is factorized as $p(\mathbf{x}_s|\mathbf{x}_t) = \prod_{i=1}^{L} p(\mathbf{x}_s^i|\mathbf{x}_t)$, where the conditional distribution for each token is:

$$p(\mathbf{x}_s^i|\mathbf{x}_t) = \begin{cases} 1, & \mathbf{x}_t^i \neq \mathbf{M}, \mathbf{x}_s^i = \mathbf{x}_t^i, \\ s/t, & \mathbf{x}_t^i = \mathbf{M}, \mathbf{x}_s^i = \mathbf{M}, \\ \frac{t-s}{t} p_\theta(\mathbf{x}_s^i|\mathbf{x}_t), & \mathbf{x}_t^i = \mathbf{M}, \mathbf{x}_s^i \neq \mathbf{M}. \end{cases} \quad (7)$$

From the above distribution, it can be seen that during the normal generation process, when $\mathbf{x}_t^i \neq \mathbf{M}$, that is, the current position has been denoised in previous steps, the token no longer changes, $\mathbf{x}_s^i = \mathbf{x}_t^i$; when $\mathbf{x}_t^i = \mathbf{M}$, the model uses two pseudo-random numbers $\mathbf{r}_\mathbf{M}^i$ and $\mathbf{r}_s^i$ generated by the PRNG to determine $\mathbf{x}_s^i$:

$$\mathbf{x}_s^i = \begin{cases} \mathbf{M}, & \mathbf{r}_\mathbf{M}^i < s/t, \\ Sample_{p_\theta(\cdot|\mathbf{x}_t)}(\mathbf{r}_s^i), & \mathbf{r}_\mathbf{M}^i \geq s/t, \end{cases} \quad (8)$$

where $\mathbf{r}_\mathbf{M}^i$ is used to determine whether the state at the current position $i$ is "keep mask" or "remove mask", while $\mathbf{r}_s^i$ is used to decide whether to sample a specific text token from the category distribution $p_\theta$ provided by the model. We denote the pseudo-random sampling process of sampling a token $x$ from the discrete distribution $p_\theta$ using the pseudo-random number $r$ as $x \leftarrow Sample_{p_\theta}(r)$.

Let the number of tokens been unmasked in step $s$ of the reverse denoising process as $N_{\text{unmask},s}$, it can be calculated as $N_{\text{unmask},s} = \sum_{i=1}^{L} \mathbb{1}\left[\mathbf{x}_t^i = \mathbf{M}\right] \cdot \mathbb{1}\left[\mathbf{r}_\mathbf{M}^i \geq s/t\right]$. These tokens will be converted in parallel and independently from masks to specific linguistic tokens within a single time step, and it can be denoted as:

$$\mathbf{x}_s^i = Sample_{p_\theta(\cdot|\mathbf{x}_t)}(\mathbf{r}_s^i). \quad (9)$$

It can be seen that the sampling process of the aforementioned diffusion language model can be divided into two stages, each controlled by a sequence of pseudo-random numbers. The first stage is determined by $\mathbf{r}_\mathbf{M}$ to identify which positions will be denoised at the current time step, while the second stage is controlled by $\mathbf{r}_s$, which determines which specific text tokens the masked positions will be predicted as.

## 4.2 Message-driven Pseudo-random Number Sampling

During steganographic embedding, we keep the first part of pseudo-random-unmasking unchanged, and adopt a message-driven PRN sampling with a fixed embedding capacity to substitute the sampling function in Formula (9). Formally, for a distribution $p_\theta(\mathbf{x}_s^j | \mathbf{x}_t)$ for a position $j$ that will be unmasked during this step, where $j \in \{j_1, \ldots, j_{N_{\text{unmask},s}}\}$, in step $t$ to $s$, if the embedding capacity is $\ell$, given message bits $\mathbf{m}^j \in \{0,1\}^\ell$ for this position, the steganographically sampled token (stegotoken) driven by the message is:

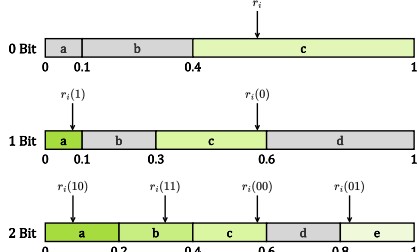

$$\mathbf{x}_s^j = Sample_{p_\theta(\cdot | \mathbf{x}_t)}(\mathbf{r}_s^j(\mathbf{m}^j)), \tag{10}$$

Figure 2: An example of message-driven PRN sampling with different capacity. Tokens that are not selected under the given PRN are marked in gray.

where $\mathbf{r}_s^j(\mathbf{m}^j) = [\mathbf{r}_s^j + \frac{\text{dec}(\mathbf{m}^j)}{2^\ell}] \mod 1$, which is a message-driven PRN, $\text{dec}(\mathbf{m})$ denotes for the decimal representation of $\mathbf{m}$. Especially, when $\ell = 0$, message-driven PRN sampling is equivalent to direct pseudo-random sampling. The algorithm for determining sampled tokens based on message-driven PRNs is presented in Algorithm 1. The corresponding algorithm for extracting secret messages based on distribution and given tokens is Algorithm 2. Both are shown in Appendix A.

Figure 2 shows how to use the embedding algorithm to embed messages into distributions with different capacities. Taking the distribution in the third row as an example, given an origin pseudo-random number , if the message to be embedded is "01", the token "e" will be selected. Similarly, during the extraction process, the message can be obtained based on the token and the PRN.

It is noteworthy that the marks that can be selected by the message after the given random number are limited. If an attempt is made to extract a message from a mark that cannot be selected, the extraction will fail. For example, the mark "d" in the third row of the figure does not correspond to any message. We call the characteristic of steganography based on message-driven pseudo-random number sampling that can only extract messages from specific tokens its **error localization property**.

### 4.3 Robust Position Embedding with Repetitive Error Correction Coding

In robust provable security steganography, token tampering differs from traditional bit flipping: each token may carry multiple bits, and tampering may cause all bits carried by the token to appear as random errors. Moreover, due to the dependence of the embedding and extraction algorithms on the model-predicted categorical distribution, incomplete token recovery will interfere with the subsequent sampling distribution, thereby invalidating message extraction.

Due to the independence of the denoising positions at the same time step, we have the opportunity to apply an error-correcting code (ECC) during the embedding process. When choosing an ECC, we hope to maintain the independence of the distribution as much as possible to reduce the entanglement between simultaneously sampled tokens; in addition, since the capacity of the secure steganography method is limited by information entropy, the code length is quite short. Formally, during the reverse denoising process from time $t$ to $s$, where positions $j_1, \ldots, j_{N_{\text{unmask},s}}$ are unmasked, a message of length $\ell_s$ is embedded into these positions using the message-driven pseudo-random steganography algorithm. We must select $\ell_s$ for all these positions in each denoising step.

Firstly, any stegosystem must meet the requirement of correctness. It can be seen from Figure 2 that if 2-bit messages are embedded in the distribution of the first row based on the given pesudo-random number, the same token "c" will be sampled using messages "00" and "01". As a result, conflicts will occur when extracting the messages. For each independent distribution $p_\theta$, the maximum capacity that can be embedded in a distribution without causing message conflicts depends on the negative logarithm of the probability of the most likely outcome, i.e., the min-entropy. Discop [16] and SparSamp [17] respectively provided solutions to the decoding uniqueness problem in message-driven random number sampling steganography systems, while also striving to maximize entropy utilization. However, their embedding schemes all require the use of multi-step joint distributions, which conflicts with our original intention of error correction for independent embedding.

Therefore, considering the trade-off between error-correcting capability and embedding capacity, to prioritize the strongest robustness, a repetition code was chosen in our method, that is, a same message $\mathbf{m} \in \{0,1\}^{\ell_s}$ is embedded for all $j \in \{j_1, \ldots, j_{N_{\text{unmask},s}}\}$ with the same length $\ell_s$ for step $s$.

For $j \in \{j_1, \ldots, j_{N_{\text{unmask},s}}\}$, the maximum embedding capacity is determined by the minimum $\ell$ value corresponding to all these positions' distributions to satisfy the min-entropy constraint. Besides, due to the requirement of repetition codes, we only embed secret messages in steps that meet the condition $N_{\text{unmask},s} \geq 3$. According to the aforementioned conditions, set

$$\ell_s = \begin{cases} \min_{j \in \{j_1, \ldots, j_{N_{\text{unmask},s}}\}} \left( \left\lfloor -\log_2 \left( \max \left( p_\theta(\mathbf{x}_s^j | \mathbf{x}_t) \right) \right) \right\rfloor \right), & N_{\text{unmask},s} \geq 3, \\ 0, & N_{\text{unmask},s} < 3. \end{cases} \quad (11)$$

which means that, for steps where $N_{\text{unmask},s}$ is insufficient or the minimum entropy is not enough, no message is involved in the generation of tokens.

We refer to positions that meet these two conditions as **robust positions**. Robust positions are a subset of denoising positions and satisfy the following: (1) The number of such positions in each step is at least 3; (2) The minimum entropy of the distribution allows each position to embed at least 1 bit. When the denoising position set of a denoising step meets the above conditions, STEAD will embed the same message bits at these positions. We denote the number of robust positions as $N_{\text{robust},s}$, abbreviated as $N_s$ (used to distinguish from the number of general denoising locations, $N_{\text{unmask},s}$).

For non-robust positions, sampling is entirely guided by PRNs $\mathbf{r}_s$. Since the sender and receiver can fully synchronize the pseudo-random number generator and the diffusion language model, the pseudo-random numbers here can serve as pseudo-random error-correcting codes [37], to cope with possible token substitutions at these non-robust positions.

### 4.4 Pseudo-random Error Correction and Neighborhood Search Extraction

During extraction, since the receiver shares the same model, PRNG, sampling settings, and key with the sender, the denoising process in the extraction process can be fully synchronized with the embedding process. For the receiver, all text positions can be divided into two categories: one is the robust position that embeds the message, and the other is the non-robust position that uses pseudo-random number sampling without embedding the message. Although the message is only embedded in the robust position, due to the distribution dependency between the preceding and succeeding time steps, the receiver needs to recover from all possible errors at all positions, rather than just dealing with the robust positions.

Firstly, we only consider token substitution as the tampering, that is, each token maintains its position in the tampered sequence as it was during generation. For each batch of robust positions, the receiver first applies the message-driven pseudo-random sampling algorithm to each position to extract the message embedded within the tokens. The intended message bits are then obtained by decoding the result with the repetition code. Correct message recovery is guaranteed as long as the number of tampered tokens in the batch does not exceed half. Subsequently, the receiver can re-run the message embedding process using the recovered message to restore any altered tokens. For non-robust positions, the pseudo-random numbers used for sampling are shared between the sender and receiver, allowing them to function as a pseudo-random error-correcting code. Consequently, the receiver can detect if a non-robust position has been tampered with by observing the outcome of the pseudo-random sampling and can readily restore the original token.

However, if errors such as insertion or deletion occur, it may cause the position of the stegotoken to be misaligned with its original sampling position during embedding. In severe cases, it may affect the message extraction of all tokens in a batch of robust positions. As shown in Figure 3, an inserted token "little" causes the original tokens at the denoising positions to shift a certain distance from their supposed positions. To address this, we have designed a $\mu$-nearest neighbor search mechanism to handle the extraction difficulties caused by the increase or decrease in token quantity. In particular, when an error occurs during extraction, we search the

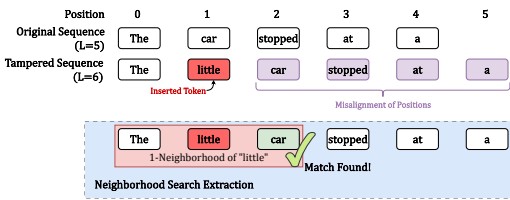

Figure 3: An example of positional misalignment due to an inserted token.

$\mu$-neighborhood corresponding to the current position, attempting to use neighborhood tokens to correct the offset of the current position. The window size $\mu$ adjusts dynamically according to the

length of the tampered text: $\mu = \max(2, |L - L'|)$, where $L$ represents the length of the original sequence, and $L'$ represents the length after tampering. This strategy adapts to different attack strengths while avoiding excessive computational complexity.

## 4.5 Proof of Security and Robustness

The security of STEAD comes from the fact that it only replaces the sampling function within each timestep of the diffusion language model during the steganographic embedding, without damaging the model's predicted probability distribution. Below, we prove that the stegotext output by the steganographic embedding algorithm and the original output randomly sampled by the model are computationally indistinguishable.

**Theorem 4.1.** *For any polynomial-time distinguisher $\mathcal{A}$, it is computationally infeasible to distinguish between the stegotext of the steganographic encoding algorithm $\mathcal{E}$ and the output of the original model sampler $\mathcal{O}$.*

**Lemma 4.2.** *For any polynomial-time distinguisher $\mathcal{A}$, its advantage in distinguishing between the output of $\mathcal{E}$ and $\mathcal{O}$ can be reduced to an advantage in distinguishing between $\mathbf{r}(\mathbf{m})$ and $\mathbf{r}$.*

*Proof.* From Formula (9) and Formula (10), it follows that at each timestep $s$, the only difference between an original sampled token and one from the steganographic encoding algorithm is the pseudo-random number used: $\mathbf{r}_s^j$ versus $\mathbf{r}_s^j(\mathbf{m})$. Since both procedures use the same PRNG as the original diffusion model, we derive Lemma 4.2.

Suppose, for contradiction, that there exists a PPT distinguisher $\mathcal{A}$ that can distinguish between PRNs after "offset" and pure PRNs with a non-negligible advantage $\epsilon(\lambda)$ such that $|\Pr[\mathcal{A}(\mathbf{r}(\mathbf{m})) = 1] - \Pr[\mathcal{A}(\mathbf{r}) = 1]| = \epsilon(\lambda)$, we use $\mathcal{A}$ to construct a new algorithm $\mathcal{A}'$, which can distinguish between PRNG outputs and truly uniform bits. On input challenge $X \in \{0,1\}^{m(\lambda)}$, $\mathcal{A}'$ answers $\mathcal{A}$'s oracle queries by $Y = \left[X + \frac{\mathrm{dec}(\mathbf{m})}{2^\ell}\right] \bmod 1$, and outputs whatever $\mathcal{A}$ outputs.

If the challenge is $X \leftarrow PRNG(\mathbf{k})$, then $\mathcal{A}'$ gives $\mathcal{A}$ exactly $\mathbf{r}(\mathbf{m})$, at this time $\Pr[\mathcal{A}' = 1 | X \leftarrow PRNG(\mathbf{k})] = \Pr[\mathcal{A}(\mathbf{r}(\mathbf{m})) = 1]$. If the challenge $X$ is uniformly random, then $\left[X + \frac{\mathrm{dec}(\mathbf{m})}{2^\ell}\right] \bmod 1$ is still uniformly random (constant offset does not change the uniform distribution), and $\Pr[\mathcal{A}' = 1 | X \leftarrow U_{m(\lambda)}] = \Pr[\mathcal{A}(\mathbf{r}) = 1]$. Then we have $|\Pr[\mathcal{A}'(PRNG(\mathbf{k})) = 1] - \Pr[\mathcal{A}'(\mathbf{u}) = 1]| = |\Pr[\mathcal{A}(\mathbf{r}(\mathbf{m})) = 1] - \Pr[\mathcal{A}(\mathbf{r}) = 1]| = \epsilon(\lambda)$, so $\mathcal{A}'$ distinguishes $PRNG(\mathbf{k})$ from uniform $\mathbf{u} \leftarrow U_{m(\lambda)}$ with non-negligible advantage $\epsilon(\lambda)$, contradicting the definition of PRNG.

Therefore, the stegotext output by the steganographic embedding algorithm and the original output randomly sampled by the model are computationally indistinguishable. $\square$

**Theorem 4.3** ($\mathcal{F}_{\alpha,\beta,\gamma}$-Robustness)**.** *Let the stego length be L. For every step $0 \le s < t \le 1$, if the adversary's tampering capabilities are bounded by $\mathcal{F}_{\alpha,\beta,\gamma}$ that satisfy $2(\alpha + \beta + \gamma) < \min_s \frac{N_s}{L}, \beta + \gamma < \frac{\mu}{L}$, where $N_s$ is the number of robust positions in time s, a STEAD stegosystem $\mathcal{S} = (\mathcal{K}, \mathcal{E}, \mathcal{D})$ with a denoising probability $\frac{t-s}{t}$ and $\mu$-neighborhood searching is robust.*

Proof of Theorem 4.3 is given in Appendix B. It is worth noting that all the discussions about tampering above are conducted at token-level. Other advanced forms of perturbation, such as tokenization ambiguity (see Appendix C) or synonym substitution, can ultimately be reduced to combinations of token-level substitution, insertion, and deletion.

## 5 Experiments

Detailed experimental settings are provided in Appendix D.

**Capacity.** In Table 1, we have calculated the effective embedding capacity of STEAD under different top-$p$ settings while also displaying the model's average entropy. We compared the embedding capacity with the latest secure robust linguistic steganography method, PSARS [23], which is based on ARM. It can be seen that under the same sampling parameters, the steganographic embedding capacity of STEAD is still significantly higher than that of PSARS (whose secure parameter is set to 32 due to a trade-off between capacity and robustness).

Table 1: Comparison of capacity and overhead of provably secure robust steganography methods

| Method | Model | Temperature | Top-$p$ | Embedding capacity (bit / $10^3$ token) | Entropy (bit / token) | Encoding rate (s / bit) | Decoding rate (s / bit) |
|---|---|---|---|---|---|---|---|
| PSARS [23] | QWEN2 | 1.0 | 1.00 | 13.81 | 3.48 | 1.6583 | 1.5893 |
| STEAD | DREAM | 1.0 | 1.00 | 84.08 | 7.78 | 0.9859 | 1.0712 |
| | | | 0.98 | 58.31 | 6.54 | 1.4431 | 1.4949 |
| | | | 0.92 | 36.33 | 4.59 | 2.3471 | 2.2642 |
| | | | 0.90 | 33.23 | 4.20 | 2.6030 | 2.3012 |

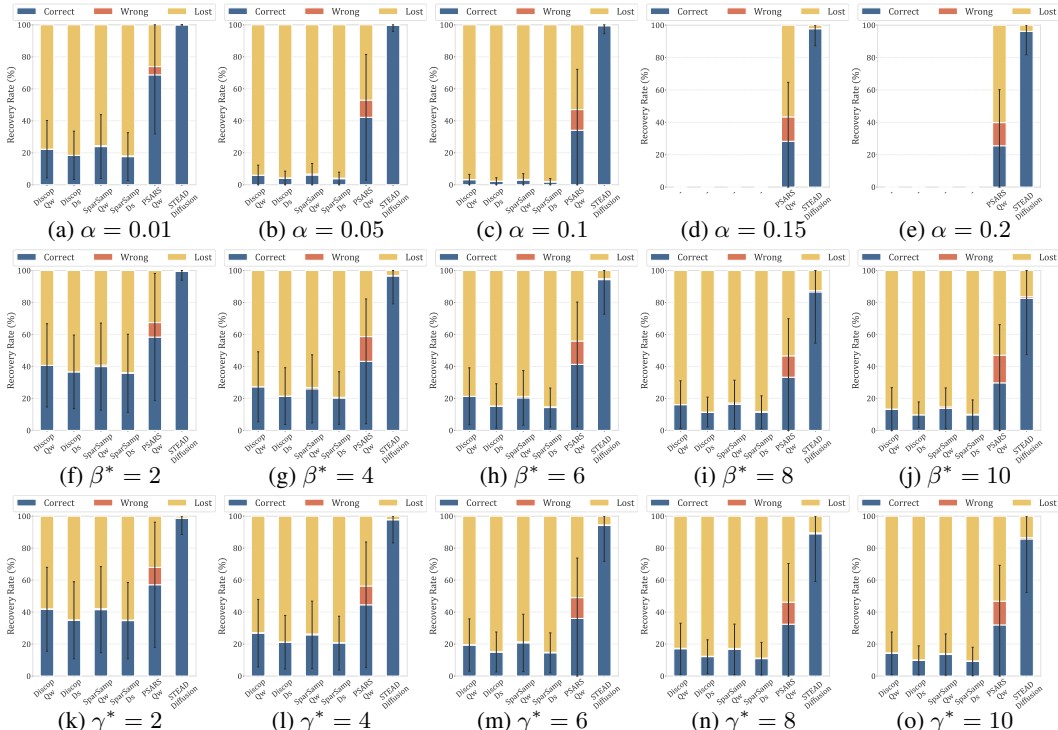

Figure 4: Robustness against token substitution, insertion and deletion. $\alpha$ represents the proportion of random substitutions, $\beta^*$ and $\gamma^*$ represent the number of random insertions and deletions. Error bars represent the standard deviation of the correct rate.

**Linguistic quality.** Figure 5 shows the average perplexity (PPL) values of stegotexts generated by STEAD and covertexts randomly sampled by the same DLM, and it can be seen that under different top-$p$ truncation settings, the PPL of the stegotexts remains consistent with the covertext.

**Statistical security.** We conducted steganalysis tests using various DNN-based steganalyzers. These tests are designed to distinguish between covertext generated from the DLM via random sampling and stegotext generated by STEAD. We generated 1000 pairs of covertext and stegotext with $p = 0.9$ under two datasets. We adopted three steganalysis methods based on deep learning: FCN [6], R-

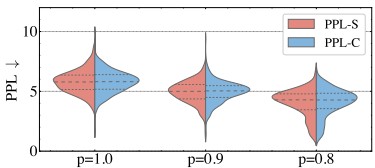

Figure 5: Comparison of PPL.

Table 2: Steganalysis results for STEAD

| Method / Dataset | FCN | R-BiLSTM-C | LSTMATT |
|---|---|---|---|
| IMDB | 50.67±0.82% | 49.00±1.59% | 49.50±1.27% |
| C4 | 49.92±2.79% | 49.08±2.66% | 50.25±0.54% |

BiLSTM-C [7], and LSTMATT [38]. Table 2 shows that the detection error rate $P_E$ approaches $50\%$. This indicates that steganalysis methods cannot perform better than random guessing in detecting stegotext generated by STEAD, which demonstrates the security of our stegosystem.

**Robustness against token-level attacks.** We apply random token-level substitution, insertion, and deletion to stegotexts, each with various intensities. The results are shown in Figure 4. Then we define mixed token-level attacks as attacks that simultaneously apply token-level substitution, insertion, and deletion. We define two attack intensities, weak and strong, using the parameter sets $(\alpha = 0.01, \beta^* = 1, \gamma^* = 1)$ and $(\alpha = 0.1, \beta^* = 3, \gamma^* = 3)$, respectively, as shown in Figure 6. It can be seen that at the token level, STEAD is more resistant to various attacks than the comparison methods, whether against a single type of attack or mixed attacks.

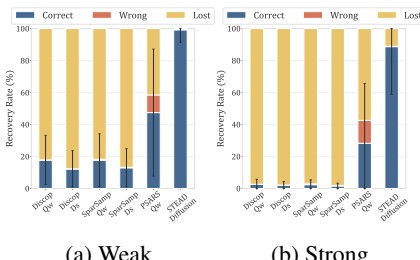

(a) Weak     (b) Strong

Figure 6: Robustness against mixed token attacks under two intensities.

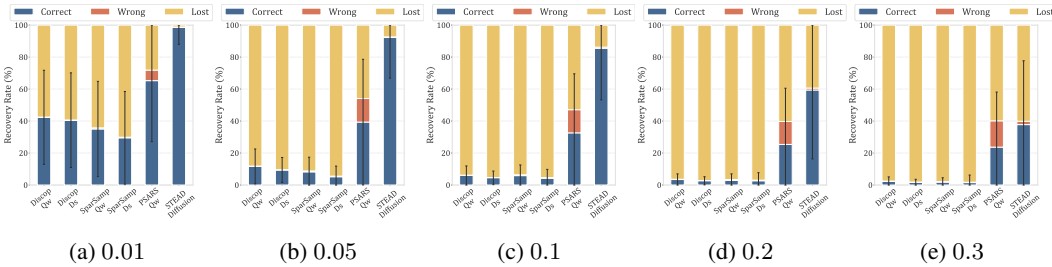

(a) 0.01          (b) 0.05          (c) 0.1          (d) 0.2          (e) 0.3

Figure 7: Robustness against semantic attacks. From left to right, substitute the synonyms of words (not tokens) in the text with proportions of 0.01, 0.05, 0.1, 0.2, and 0.3 respectively.

**Robustness against realistic attack scenarios.** Figure 7 shows the robustness evaluation to more challenging scenarios. There is a semantic synonym substitution attack at word-level based on TextAttack [39]. With a word substitution rate of 0.1, non-robust methods are rendered almost entirely ineffective, whereas STEAD sustains an extraction correction rate above $80\%$.

**Ablation study.** Our method comprises three key components: a message-driven PRN sampling algorithm, robust position embedding with error correction coding (RPE+ECC), and a neighborhood search extraction (NSE) strategy. The ablation study for these components is presented in Figure 8. We evaluated robustness against token substitution (using $\alpha = 0.2$), token insertion (with $\beta^* = 10$), and token deletion

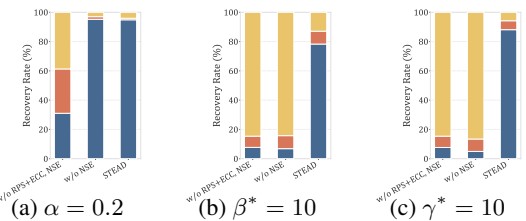

(a) $\alpha = 0.2$     (b) $\beta^* = 10$     (c) $\gamma^* = 10$

Figure 8: Ablation study.

(with $\gamma^* = 10$). The baseline steganographic algorithm (i.e., without RPE, ECC, and NSE) performs embedding and extraction directly on the DLM. Its robustness is comparable to that of the ARM-based method, indicating that the advantages of STEAD are not derived solely from the DLM itself. The integration of RPE and ECC confers resistance to substitution by leveraging the DLM's parallel characteristic at specific, limited positions—a key innovation of STEAD. Furthermore, the inclusion of NSE enables STEAD to handle insertion and deletion.

## 6  Conclusion

We propose STEAD, a novel, robust, and provably secure linguistic steganography method utilizing a Diffusion Language Model (DLM). STEAD employs the random synchronous sampling characteristic of the DLM, combined with error correction codes and a neighbor search extraction strategy, to achieve robustness against substitution, insertion, deletion, and token ambiguity. We demonstrate the performance of STEAD through theoretical proofs and extensive experiments. Results indicate that STEAD exhibits a higher embedding capacity and enhanced robustness over comparable methods.

## Acknolegment

This work was supported in part by the National Natural Science Foundation of China under Grant U2336206, Grant 62472398, Grant U2436601, and Grant 62402469. We are grateful to Bai et al., the authors of [23], for generously providing the code associated with their work and offering valuable guidance.

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

# A  Algorithms of STEAD

## A.1  Message-Driven Pseudo-Random Number Sampling

See the Algorithm 1.

## A.2  The Main Process of Message Extraction

In this section, we describe the main process of message extraction for the receiver in the STEAD stegosystem. Also, we give in detail the error correction and $\mu$-neighborhood search strategy.

For each stego, the receiver initializes an offset vector $\mathbf{o}^{1:L}$ with all zeros, $\mathbf{o}_{t=1}^{1:L} = 0$. During the extraction process at each time step $t \rightarrow s$, the receiver first checks the denoising positions $\{j_1, \ldots, j_{N_s}\}$ by $\mathbf{r_M}$. Then the receiver finds the corresponding stegotokens $\mathbf{st}^{j_1, \ldots, j_{N_s}}$. Calculate the corresponding stego capacity $\ell$ at the denoising positions by Formula (11) based on the predicted distribution $p_\theta(\mathbf{x}_s^j|\mathbf{x}_t)$. If $\ell = 0$, the receiver can directly check whether the stegotoken $\mathbf{st}^j$ at each denoising position $j$ has been tampered with using the PRN $\mathbf{r}_s^j$. If tampered, the corrected token can be obtained by Formula (9). If $\ell > 0$, for each denoising position $j$, the receiver attempts to extract the secret message $\mathbf{m}_s$ using the distribution $p_\theta(\mathbf{x}_s^j|\mathbf{x}_t)$, the corresponding stegotoken $\mathbf{st}^j$, and the

**Algorithm 1** EMBED($P, r, \mathbf{m}$): an message embedding algorithm based on message-driven pseudo-random number sampling

---

    **Input:** A distribution $P$, a pseudo-random number $r$, optional $\ell$-bit message $\mathbf{m}$
1:  $cumul \leftarrow 0$
2:  **if** $\ell > 0$ **then**
3:     $r \leftarrow \left[ r + \frac{\text{dec}(\mathbf{m})}{2^\ell} \right] \mod 1$
4:  **end if**
5:  **for** $k \in \{0, 1, \ldots, |P|\}$ **do**
6:     $cumul \leftarrow cumul + P(k)$
7:     **if** $cumul > r$ **then**
8:         $x \leftarrow$ token corresponding to $P(k)$
9:         **break**
10:    **end if**
11: **end for**
    **Output:** sampled token $v$

---

**Algorithm 2** EXTRACT($P, r, x$): an message extracting algorithm based on message-driven pseudo-random number sampling

---

    **Input:** A distribution $P$, a pseudo-random number $r$, a sampled token $x$
1:  $cumul, r_{left}, r_{right} \leftarrow 0$
2:  **for** $k \in \{0, 1, \ldots, |P|\}$ **do**
3:     $cumul \leftarrow cumul + P(k)$
4:     **if** x corresponds to P(k) **then**
5:         $r_{left} \leftarrow cumul - P(k)$
6:         $r_{right} \leftarrow cumul$
7:         **break**
8:     **end if**
9:  **end for**
10: **for** $m \in \{0, 1, \ldots, 2^\ell\}$ **do**
11:    **if** $r_{left} < \left[ r + \frac{\text{dec}(\mathbf{m})}{2^\ell} \right] \mod 1 < r_{right}$ **then**
12:       $\mathbf{m} \leftarrow \text{bin}(m)$                                   ▷ Succeed to extract
13:       **return** $\mathbf{m}$
14:    **end if**
15: **end for**
16: **return** $\mathbf{m} \leftarrow$ "x"                                                    ▷ Fail to extract
    **Output:** Extracted message $\mathbf{m}$

---

PRN $\mathbf{r}_s^j$ using Algorithm 2. Since the secret messages embedded at all denoising positions of step $t \rightarrow s$ are encoded using a repetition code, the receiver can decode the most likely embedded secret message fragment at the current time step through voting and identify which positions may have been tampered with. The receiver can then correct the erroneous positions using the decoded message.

However, due to the possible increase and deletion of tokens, there may be a situation where all the stegotokens at the denoising positions are shifted, making it impossible to extract the correct message from any of the positions.

For all positions where extraction fails, the receiver checks whether they can be corrected using the existing offset. If the correction is successful, the stego is updated, and the process proceeds to the next position. If the offset correction is ineffective, or the offset value at the current position is zero, the receiver performs a neighborhood search. This involves scanning for tokens within the range of $\mu$ around the current position of the stegotoken in the stego sequence. For each neighboring token, an attempt is made to extract the information. If a token that can be successfully extracted is found during the neighborhood search, it is considered that the correct stego position has been found. The offset relative to the original denoising position is calculated, and the offset vector is updated from the current position to the next non-mask position. Finally, the correction of the generation process is completed.

## B  Proof of Robustness

*Proof.* We divide failure into two independent cases and bound each probability separately.

(1) Since the repetitive code has a minimum distance $N_s$, up to $\lfloor N_s/2 \rfloor$ token substitution can be corrected. The adversary can replace at most $\alpha L$ tokens total. In one step, $N_s$ tokens out of $L$ are denoised, chosen uniformly at random. By a Chernoff bound,

$$\Pr\big(\text{substitutions in step} > N_s/2\big) \leq \exp\Big(-\frac{(N_s/2 - \alpha L)^2}{2\alpha L}\Big). \tag{12}$$

Since $2\alpha < N_s/L$, this exponent grows with $L$, making the probability negligible. Over polynomially many diffusion steps, the total remains negligible.

(2) Insertions or deletions shift stego positions by at most $(\beta + \gamma)L$. During extraction, for each expected position $i$, we search within its $\mu$-neighborhood $\{i - \mu, \ldots, i + \mu\}$ for the most likely token under the original distribution. If $(\beta + \gamma)L < \mu$, then even after insertions and deletions, every remaining stego token remains within the search window. Thus, all positions can be realigned correctly. Since addition and deletion also mean that the token at position $i$ changes. Therefore, it is also necessary to add $\beta$ and $\gamma$ to the previously calculated ECC boundaries.

Combining these results, the STEAD stegosystem can achieve robustness against $\mathcal{F}_{\alpha,\beta,\gamma}$-bounded adversaries, when $2(\alpha + \beta + \gamma) < \min_s(N_s/L)$, and $\beta + \gamma < \frac{\mu}{L}$. $\qquad\square$

## C  Deal with Tokenization Ambiguity

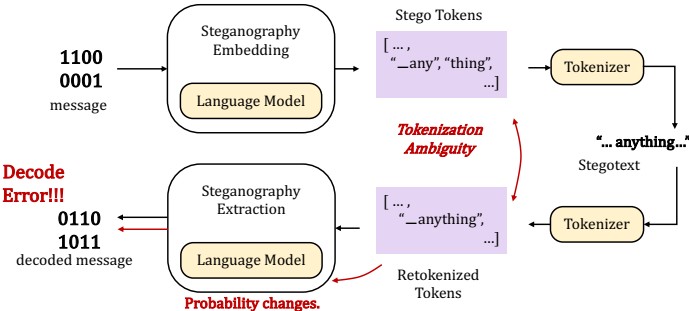

Figure 9: An example of tokenization ambiguity given by [28]. The sender generates a token sequence corresponding to subwords "_any" and "thing" during steganography embedding. During transmission, the stego-tokens are decoded into the text "anything". Unfortunately, the receiver may retokenize "anything" as a single token "_anything". This can lead to errors in steganography extraction.

Tokenization ambiguity (TA) [27, 29, 30, 28, 17], or segmentation ambiguity, refers to a phenomenon that a tokenizer may not re-decode a token sequence into the same tokens. Figure 9 gives an example. It can be seen that the appearance of TA is basically accompanied by the increase or decrease in the total number of tokens, as well as changes in specific tokens. Although there are existing methods that do not harm the distribution to eliminate token ambiguity in steganography, such as backtracking with checkpoints [40] and SyncPool [28], these methods do not constitute a competitive relationship with the method proposed in this paper. The impact of TA will not exceed the limited threat model of substitution, insertion, and deletion discussed in Section 3.2. Therefore, the proposed provably secure robust steganography method in this paper can naturally handle most token ambiguity cases, unless the ambiguity phenomenon occurs too frequently. However, the method aiming at simply eliminating ambiguity cannot counter the active tampering of the adversary.

# D  Experimental Settings

## D.1  Models and datasets.

We adopt the latest advanced text diffusion model Dream [36] as the stego generator. For the autoregressive model, we choose two popular models, Qwen2.5-7B [41] and Deepseek-7B-base [42], with similar performance to Dream to fairly compare the performance of steganographic methods. We randomly select 200 text from the IMDb dataset [43] as input for the text generation task. Each text is truncated to the first two sentences for context. The model generates 512 tokens for each context as stego or cover under specific sampling settings.

## D.2  Configs.

In the text sampling process of generating text by language models, text quality is influenced by three sampling parameters: **temperature**, **top-p**, and **top-k**. The temperature controls the randomness of the output. The lower the **temperature**, the more deterministic the text is (which usually leads better quality); versa, the more diverse it is. Top-p sampling (a.k.a. Nucleus Sampling) dynamically selects the smallest set of words with cumulative probability exceeding the threshold $p$. Top-k sampling samples from the highest probability $k$ words at each step. The three together adjust the deterministic and diversity of the generated text. We test our method under various temperature, top-p, and top-k settings.

## D.3  Metrics.

There are four evaluation aspects in the experiment, namely **capacity**, **security**, and **robustness**.

**Capacity** includes effective capacity and theoretical limit capacity, respectively represented as **embedding capacity** and **entropy**. The embedding capacity is calculated as the average number of bits per stego token embedded. We calculate the sum of the entropy of the probability distribution of denoised tokens at each step in the process of generating the complete token sequence by the model. The sum of the entropy divided by the total number of tokens is the theoretical limit of the embedding bit number per token.

**Security** is evaluated through three metrics: **Kullback-Leibler divergence (KLD)**, $P_E$, and **perplexity (PPL)**. **KLD** measures the distribution difference between cover and stego from the perspective of information theory [44]. It is calculated as the relative entropy between the probability distribution of generating cover and that of generating stego. $P_E$ represents the detection error rate of a steganalyzer, which is calculated as the proportion of samples misclassified to the total number of samples. It is used to empirically demonstrate the ability of the stego bypass a detector. PPL reflects the uncertainty of the model in text prediction. The lower the PPL, the better the model. By comparing the PPL changes of the stego and cover generated by the model, it can also reflect the degree of influence of the steganographic methodon the statistical distribution of the generated text.

**Robustness** is evaluated through: (1) a coarse-grained metric, **success rate** of message extraction which is calculated as the proportion of samples where the message is completely correctly extracted (binary outcome), and (2) a set of fine-grained metric, the **recovery rate** of message denoted as a triplet (**correct rate**, **wrong rate**, **lost rate**). Specially, when stego is perturbed, the extraction algorithm may not be able to extract a message of the same length as the embedded message, which means that the message after a certain bit has been lost. We calculate the lost rate as $1 - \min(\ell^{ext}, \ell^{emb})/\ell^{emb}$ where $\ell^{ext}$ and $\ell^{emb}$ are the length of extracted message and embedded message. Correspondingly, the correct rate and wrong rate represent the proportion of correct bits (of non-lost message) and wrong bits (of non-lost message), respectively, among the total embedded message bits.

# E  Additional Experiments

## E.1  Migration and Redundancy Experiments

### E.1.1  Failure of Migrating Existing Methods to Diffusion Models

Although text diffusion models provide opportunities for designing robustness in provably secure linguistic steganography, simply migrating existing methods based on the autoregressive model to diffusion models is still infeasible. We directly replace the random sampling algorithm in the generation process of the diffusion model with the existing steganographic algorithm, achieving a simple migration from ARM version to DLM version. Table 3 demonstrates the success rate of message extraction of the ARM version of Discop, the DLM version of Discop, and our method STEAD under token ambiguity. The results show that direct migration has seriously damaged the availability of Discop.

Table 3: Results of Migrating Existing ARMs-based Methods to Diffusion Models

| Method/ Model | Discop/ Qwen | Discop/ DeepSeek | Discop/ Dream | STEAD/ Dream |
|---|---|---|---|---|
| Correct rate (%) | 96.00 | 98.82 | 0.94 | 99.49 |
| Wrong rate (%) | 0.00 | 0.00 | 0.65 | 0.00 |
| Lost rate (%) | 4.00 | 1.17 | 98.41 | 0.51 |
| Success rate (%) | 95.50 | 97.50 | 0.00 $\downarrow$ | **99.49** |

### E.1.2  Failure of Adding Redundant Encoding to Existing Methods

Adding redundancy codes to existing provably secure steganographic methods based on autoregressive models may seem like a straightforward way to enhance robustness. However, ARMs suffer from error propagation—tampering with one token disrupts the probability distribution of all subsequent sampling steps (see Figure 1). Therefore, using error-correcting codes (e.g., repetition codes, BCH, or LDPC) cannot make this idea of directly combining non-robust methods with redundancy encoding feasible. To demonstrate this limitation, we take the Qwen model as an example and apply a $(20, 1)$ repetition code to the Discop method. The experimental results of robustness agaisnt mix attack (with strong intensity) in Table 4 confirm this claim.

Table 4: Results of applying a (20,1) repetition code to Discop on Qwen.

| Method/ Model | Capacity (bit) | Correct rate (%) | Wrong rate (%) | Lost rate (%) | Decoded length (bit) |
|---|---|---|---|---|---|
| Discop/ Qwen | 59.70 | 2.55 | 0.00 | 97.45 | 1.21 |
| STEAD/ Dream | 23.93 | 83.43 | 0.79 | 15.78 | 20.06 |

## E.2  Additional Results about Ablation Study

### E.2.1  Why Use Repetition Codes

The use of repetition codes in STEAD is motivated by the unique characteristics of robust provably secure text steganography:

**Scenario-specific requirements.**  Unlike traditional bit-flipping in lossy channels, text token tampering can corrupt multiple bits simultaneously (since each token carries multiple bits). For provably secure steganography, full token recovery is critical—any residual error propagates to subsequent extraction steps. This demands a mechanism robust to complete token-level failures.

**Constraints on code parameters.**  In STEAD's one-step generation, robust positions support very short code lengths with high error-correction demands. Table 5 shows that under typical conditions (assuming $\leq 1$ token error per step), linear codes degenerate to repetition codes when parameters approach $(n, 1)$.

**Trade-off clarification.** We acknowledge the trade-off between robustness and embedding capacity. Repetition codes were chosen to prioritize maximum robustness, which is critical for the first DLM-based provably secure method. While algebraic codes (e.g., BCH) offer efficiency in longer codes, they underperform in our short-code, high-error scenario.

Table 5: Degeneration of linear codes to repetition codes under short-length constraints.

| Dataset | IMDB | C4 |
|---|---|---|
| Code length | 6.28 | 6.69 |
| Error length | 1.92 | 2.05 |

### E.2.2 Computational Overhead of Neighborhood Search Strategies

The window size $\mu$ is dynamically adjusted based on the tampered text length: $\mu = \max(2, |L - L'|)$, where $L$ denotes the original sequence length and $L'$ is the length after tampering. This strategy balances adaptability to varying attack intensities (e.g., insertion/deletion) while avoiding excessive computational overhead. To validate this setting, we tested decoding performance under different fixed $\mu$ values using the strong mixed attack scenario. Table 6 shows that the larger $\mu$ achieves higher accuracy without significant time overhead per effective bit.

Table 6: Computational overhead of neighborhood search strategies with varying window sizes $\mu$.

| $\mu$ | Total time (s) | Decode time (s/bit) | Correct rate (%) | Wrong rate (%) | Lost rate (%) |
|---|---|---|---|---|---|
| 1 | 48.22 | 2.80 | 70.99 | 1.06 | 27.96 |
| 2 | 59.46 | 2.78 | 88.80 | 0.26 | 10.94 |
| 4 | 62.12 | 2.77 | 92.87 | 0.33 | 6.79 |

### E.3 Quantitative Estimation of Provable Robustness Boundaries

We tested the average number of occurrences of different values of $N_s$ in a 512-length token sequence generation under various prompts (using different IMDB datasets and C4 datasets). Due to the repetition code we adopt, embedding is required only when $N$ is greater than 3. As shown in Table 7, although the distribution of $N_s$ is different, the minimum value of $N_s$ is always 3.

The theoretical assumption $2(\alpha + \beta + \gamma) < \min_s(N_s/L)$ defines the boundary for perfect 100% extraction accuracy. In practice, this boundary is strict and rarely met in complex real-world scenarios. For example, in generating 512-length text, $\min N_s$ is typically 3, leading to a theoretical tampering threshold of 0.3% (i.e., 1.5 tokens out of 512). This strict boundary is difficult to satisfy under realistic attacks. However, this does not invalidate message extraction beyond the boundary.

Experimental results (Figures 4, 6, 7) show a gradual rather than abrupt decline in accuracy: (1) At low tampering intensities (below the boundary), extraction accuracy remains nearly 100%; (2) As tampering exceeds the boundary, accuracy decreases steadily but retains practical utility (e.g., 82.63% correct extraction even with 10 insertions in 500 samples, as shown in prior robustness tests). This aligns with real-world language environments, where attacks vary in intensity but rarely cause extreme tampering. The theoretical bound serves as a benchmark for optimal performance, while empirical results demonstrate the method's resilience beyond this idealized scenario.

Table 7: Empirical statistics of $N_s$ supporting the estimation of provable robustness boundaries.

| Dataset | $N_s = 3$ | $N_s = 4$ | $N_s = 5$ | $N_s = 6$ |
|---|---|---|---|---|
| IMDB | 16.2 | 3.3 | 0.6 | 0.1 |
| C4 | 13.2 | 2.8 | 0.45 | 0.15 |

### E.4 Results of Robustness under Token Ambiguity

In previous experiments, all attacks were conducted at the token level. In the steganography scenario, a more general case is that the receiver can only obtain the text. Table 8 shows the extraction success

rates of different steganography methods when directly extracting the steganographic text. All the compared methods did not add any disambiguation means, only comparing the robustness of the provably secure steganographic methods themselves against TA.

Table 8: Robustness under TA

| Method/ Model | Discop/ Qwen | Discop/ DeepSeek | SparSamp/ Qwen | SparSamp/ DeepSeek | ARS/ Qwen | STEAD/ Dream |
|---|---|---|---|---|---|---|
| Success rate | 95.50% | 97.50% | 92.42% | 96.00% | 89.04% | **99.49%** |

# F   Visualization of Generated Texts

**Example Stegotext 1**  I can understand your concern. During pregnancy, watching some sensitive or potentially graphic content, especially content about violence, gore, etc., could certainly be unsettling and disturbing. It's completely normal for watching videos or content that affected you to be cared for. Being aware of your reaction, you're absolutely not required to watch anything else that can upset you again. That's part of the privilege of understanding that pregnancy is an emotional journey too, for you.

It's generally wise to be cautious with the types of content you watch or information you consume, especially when you're trying to enjoy your pregnancy safely and comfortably. News, documentaries, and videos that might show harm to others, especially in violent situations, can cause some people severe emotional distress. Moreover, emotional reactions can be highly amplified during pregnancy, so it's good to take steps to keep your emotions centered and protected.

Here are some other things you might want to do to ensure a positive and healthy mental and physical experience during pregnancy:
– **Regular relaxation and stress management techniques**: This can include activities such as meditation, yoga, or deep breathing exercises.
– **Healthcare check-ups**: Regular visits to a healthcare provider can provide you with support regarding mental health, as well as offer guidance about nutrition and exercise, or if your blood pressure or weight need to be on track.
– **Create a support system**: Let a trusted partner or friend know about how you might be feeling. It's okay to ask for help when and if you need it.
– **Community of parents and other parents-to-be**: Sometimes, stories and experiences can help normalize the pregnancy journey. Consider a local community group or online support group to find people whose advice and encouragement can be supportive.

You're not alone in your pregnancy, and it's perfectly alright to acknowledge the need and avoid this kind of content if makes you feel upset or harmful to yourself.

Moreover, if you ever feel persistent symptoms, such as:
– A change in appetite
– Intense crying without a sign of any problem
– Irritability or persistent or severe nausea to the point to disturb your day
– Persistent, severe headaches
– Severe anxiety

Then it's appropriate to reach out for support. A doctor or healthcare professionals are ready to assist and guide you, don't let you suffer through it.

Remember, your health and happiness during pregnancy is very important, and you can take the necessary steps to make your own pregnancy experience a safe and positive one. Every pregnancy is unique, so are our needs and preferences. Good luck!

**Example Stegotext 2**  It's great to hear that you've found a movie that you can really enjoy. In the future, I hope you'll be more careful in the future when you're in the process of reviewing movies. I'm an AI and I don't keep up with trends or watch movies, but I can give you some guidance and tips on how to write a good review.

Let's imagine a movie review and pretend that it's from a movie by a director named Madsen. A major fan of Madsen films might think that this would be the type of movie you might want to avoid, if I've been recommending Madsen movies to you. In fact, I'm not sure if the movie has a Madsen movie in it (they probably don't). This might cause some of you to avoid this movie. I'll give you a tip in the description what it's about:

I know you probably love Madsen's films too. I'm a fan of his movies and I have a lot of his movies in my collection. But trust me, this is not the type of movie that you want to watch. It's a terrible movie that's bad for every reason in existence. Horrible plot holes, terrible actors, and horrendous writing. It's a tough watch that will make you sad for Madsen's movies. It's a shame, but I have to warn you, that it's not possible for you to enjoy it. However, the film has a small amount of humor of irony, that makes it funny at times. I can assure you, if you are a Madsen fan, you will not care one bit. So, please, avoid that movie. You've been warned.

When writing a review for any other movie from Madsen, take note of what feels right to you. If it's funny, or dramatic, mention some of those things in it. Don't forget to give your honest opinion and why. If there's anything that you have about a certain genre, or about a particular director, don't be afraid to point it out, but be general. Don't single out a Madsen movie or a genre. Think of a target audience for which you might be or might not be the target audience wanting to see it. This way, you'll be giving a review that will be helpful to potential viewers who share your interests. This will make your review more helpful to others who might or might not appreciate the movie as well as you found the experience be.

