# OpenReview forum: "STEAD: Robust Provably Secure Linguistic Steganography with Diffusion Language Model"
_NeurIPS.cc/2025/Conference — NeurIPS 2025 poster_

### Official Review · Reviewer_JQ4Y · 2025-06-26

**Clarity:** 2
**Significance:** 2
**Originality:** 2
**Rating:** 4
**Confidence:** 3

**Summary:**

Previous works on provably secure linguistic steganography have used autoregressive language models, which are not robust against tampering attacks. This paper proposes using diffusion language models instead, as they allow for identifying robust positions for steganographic embedding. Combined with the introduced error correction strategies, the proposed method improves robustness while maintaining security.

**Questions:**

1. What is the error positioning characteristics of message-driven PRN sampling and how to use it to judge the extraction error for replacement, insertion, and deletion?

2. What is the computational cost of neighborhood search extraction?

**Ethical Concerns:**

["NO or VERY MINOR ethics concerns only"]

**Final Justification:**

The authors make detailed plan to improve clarity and presentation. While NSE enhances robustness, it also results in significant decoding latency overhead. Therefore, I am increasing my rating to a 4, but not higher.

**Limitations:**

Not see limitations being discussed in the paper.

**Quality:**

2

**Strengths And Weaknesses:**

**Strengths**

1. The paper achieves both secure and robust linguistic steganography.

2. It considers a wide range of tampering attacks, including substitution, insertion, deletion, and token ambiguity.

**Weaknesses**

1. No ablation studies are provided to demonstrate the effectiveness of the proposed components, such as robust position selection, error correction coding (ECC), or Neighborhood Search Extraction.

2. The figures lack sufficient descriptions and explanations, which hinders understanding of the methodology.

3. The methodology section lacks clarity. Specifically, it is unclear how robust positions are selected for embedding.

4. As noted in line 104, GTSD is another work that leverages diffusion models to enhance robustness in linguistic steganography (which also demonstrates to be secure), but no experimental comparison is provided.

---

> ### Author Rebuttal · Authors · 2025-07-30
>
> Thank you for your valuable comments. Here are our responses.
>
> 1. **Weakness1: No ablation studies are provided to demonstrate the effectiveness of the proposed components, such as robust position selection, error correction coding (ECC), or Neighborhood Search Extraction.**
>
>     **Response:**
>
>     We have added ablation studies for the proposed components as suggested. Our framework is composed of a message-driven PRN sampling algorithm, **robust position selection (RPS)** and **error correction coding (ECC)**, and the **neighborhood search extraction (NSE)** strategy.
>
>     The basic steganographic algorithm (w/o RPS, ECC, NSE) directly performs steganographic embedding/extraction on the DM model. Its robustness is similar to that of the ARM-based method. Adding RPS and ECC enables resistance to token substitution tampering by leveraging DM's parallel denoising capability on limited positions—a key innovation of STEAD. Further adding NSE enhances adaptation to token offset tampering, allowing STEAD to robustly handle substitution, insertion, and deletion (see Table below).
>
>     | temperature=1.0, p=0.95 | Replacement 0.2 | Insertion 10 | Deletion 10 |
>     | --- | --- | --- | --- |
>     | w/o RPS, ECC, NSE | 31.03/30.27/38.70 | 7.73/7.67/84.60 | 7.67/7.72/84.61 |
>     | w/o NSE | 95.22/1.77/3.01 | 6.85/8.89/84.26 | 4.96/8.47/86.57 |
>     | STEAD | 94.87/1.07/4.06 | 78.31/8.76/12.92 | 88.11/6.10/5.79 |
> 2. **Weakness2: The figures lack sufficient descriptions and explanations, which hinders understanding of the methodology.**
>
>     **Response:**
>
>     Thank you for your suggestion; we will thoroughly revise all figures with more detailed descriptions and explanations to enhance methodological clarity:
>
>     1. For Figure 1, we will elaborate on error propagation mechanisms. Specifically, using the formula in Line 77, we will explicitly illustrate how corruption of a preceding word alters the probability distribution of the next word in the ARM model.
>     2. For Figure 2, a comprehensive explanation of the STEAD stegosystem will be added: STEAD is composed with message-driven Pseudo-Random sampling, Robust Position Selection (RPS), Error Correction Coding (ECC), and Neighborhood Search Extraction (NSE).
>     3. For Figure 3, three examples of distributions are given from top to bottom, corresponding to embedding 0, 1, and 2 bits of message. $r_i$ represents the initial random number, and $r_i(X)$ represents the random number corresponding to the message X.
>     4. For Figure 4, when mu=1, if the extraction of "little" fails, then traverse the two tokens "The" and "car" to try to extract valid message bits.
>     5. **Experimental figures**: Captions will explicitly state sampling parameters (temperature=1.0, p=0.95). Experimental analysis will include quantitative descriptions of trends (e.g., "compared to ARS, the correct extraction ratio of token replacement with STEAD's strength of 0.1 has increased by more than 80").
> 3. **Weakness3: The methodology section lacks clarity. Specifically, it is unclear how robust positions are selected for embedding.**
>
>     **Response:**
>
>     We would like to clarify that our method is composed of the following parts:
>
>     - Message-Driven Pseudo-Random Sampling: Secret bits are conveyed by adding a small, message-dependent offset to the pseudo-random number that drives sampling within each token distribution. Because the underlying distribution is unaltered, the stegotext remains provably indistinguishable from ordinary output.
>     - Robust Position Selection: At each reverse-diffusion step, STEAD identifies the subset of positions that are unmasked in parallel. These positions are independent of one another, making them suitable for isolated, error-tolerant embedding. When they are capable to carry enough bits, they are selected as robust positions.
>     - Error Correction Coding: Repetition codes are used to repetitively embed message across robust positions of a step, enabling correction of up to half the substitutions in that batch. Positions without payload serve as pseudo-random checks that can self-correct single-token substitutions.
>     - Neighborhood Search Extraction: A µ-neighborhood search strategy realigns the decoding process when global offsets arise from insertions or deletions.
>
>     In summary, we proposed a  stegosystem based on diffusion language models, which is simultaneously provably secure and robust.
>
>     Addressing your specific concern—*"it is unclear how robust positions are selected for embedding"*:
>
>     In order to select robust positions, it is necessary to first determine the denoising positions of each step. As clarified in lines 194–196, **denoised positions** are independently sampled via conditional probabilities (formula 7) during each step (consistent across cover and stego generation), regardless of embedding.
>
>     **Robust positions are a subset of denoised positions that meet two criteria** (formula 12):
>
>     1. The number of such positions in each step is at least 3 (to support repetition codes).
>     2. The minimum entropy of their distribution allows embedding at least 1 bit per position.
>
>     Notably, robust positions are derived from denoised positions but are not identical to them. Only those meeting the two criteria above qualify as robust positions for message embedding.
>
>     We will integrate this explanation into Section 4 to explicitly illustrate the steganographic embedding process in the diffusion model, ensuring full clarity.
>
> 4. **Weakness4: As noted in line 104, GTSD is another work that leverages diffusion models to enhance robustness in linguistic steganography (which also demonstrates to be secure), but no experimental comparison is provided.**
>
>     **Response:**
>
>     GTSD (*Submitted to arXiv on 28 Apr 2025*) is our **Contemporaneous work.** To ensure the sufficiency of the current situation described in this paper, we added relevant references, and the paper was officially published after our submission (*24 June 2025*), so we did not conduct a comparison.
>
>     Notably, **GTSD and our method (STEAD) differ fundamentally**:
>
>     - GTSD is **not provably secure (empirically secure)**. Its diffusion model requires dataset-specific training, and the embedding process introduces non-zero KL divergence, leading to detectable distribution differences (as reported, steganalysis detectors achieve ~10% accuracy above random guessing). This leaves it vulnerable to evolving steganalysis techniques.
>     - In contrast, STEAD is **provably secure**, theoretically guaranteeing indistinguishability between stego and cover data, eliminating detection risks—a key focus in current steganography research.
>
>     They are two types of methods, empirically secure and theoretically secure. So a direct quantitative comparison is not appropriate. We will add the above qualitative analysis to our revised paper.
>
> 5. **Question1: What is the error positioning characteristics of message-driven PRN sampling and how to use it to judge the extraction error for replacement, insertion, and deletion?**
>
>     **Response:**
>
>     The error positioning characteristic of message-driven PRN sampling is that the steganographic method can **only extract messages from specific tokens**, as defined in Appendix A.
>
>     This characteristic is applied to judge extraction errors for replacement, insertion, and deletion as follows:
>
>     The number of tokens selectable via the message is limited. If an unexpected token appears at a designated position, it directly indicates tampering at that position. For example, in Figure 9, tokens "a""b""c""e" correspond to messages "10""11""00""01"; the presence of "d" at this robust position confirms a token change. This serves as a sufficient (but not necessary) condition for identifying tampering.
>
>     Additionally, repetition codes play a key role: positions where the extracted message conflicts with the repetition code decoding (voting result) are flagged as potential tampering sites. Together, these two mechanisms enable the identification of replacement, insertion, and deletion errors.
>
> 6. **Question2: What is the computational cost of neighborhood search extraction?**
>
>     **Response:**
>
>     The computational cost of neighborhood search extraction is evaluated via decoding time under different window sizes (μ) against the mixed attack (a stronger attack scheme), as shown in the table below:
>
>     |  | μ=1 | μ=2 | μ=4 |
>     | --- | --- | --- | --- |
>     | decode time（s） | 48.22 | 59.46 | 62.12 |
>     | decode time（s/bit) | 2.80 | 2.78 | 2.77 |
>     | Correct | 56.85 | 82.58 | 90.58 |
>     | Wrong | 15.05 | 6.19 | 2.77 |
>     | Lost | 28.10 | 11.23 | 6.65 |
>
>     As μ increases, decoding complexity rises (longer total decode time) but yields higher extraction accuracy. Notably, the time per bit of effective data slightly decreases (from 2.80 s/bit to 2.77 s/bit).
>
>     In non-lossy environments, neighborhood search extraction is not triggered, resulting in zero additional time consumption. For all attack experiments, we adopt a dynamic window size adjustment strategy: μ is set to $max(2,|L-L'|)$, where $L'$ is the tampered token sequence.

---

> > ### Comment · Reviewer_JQ4Y · 2025-08-04
> >
> > Thanks for the rebuttal.
> >
> > The presentation and clarity of the paper still need improvement.
> >
> > Regarding my point on insufficient figure descriptions—take Figure 2 as an example. The caption simply states: `An overview of the proposed STEAD stegosystem.` In the main text, the only reference is: `Figure 2 shows the overview of STEAD.` As a result, readers are left with minimal understanding of what the figure conveys. This is problematic because Figure 2 is central to the paper and contains many important components. Imagine explaining this figure to others—just calling it an "overview" is not enough. Even in the rebuttal, stating that the figure includes "message-driven Pseudo-Random sampling, Robust Position Selection (RPS), Error Correction Coding (ECC), and Neighborhood Search Extraction (NSE)" is insufficient unless the figure clearly illustrates how these components work together.
> > In the revision, please walk the reader through the figure, either in the caption or in the main text when it is referenced, so that its contents are properly contextualized and explained.
> >
> >
> > Also, robust position and denoising position should be more clearly explained in the main text. This lack of clarity also led to confusion from reviewer QWsN. For example, in the rebuttal you state: `When they are capable to carry enough bits, they are selected as robust positions.` This is vague. In the revision, please define what qualifies as "enough" in precise and technical terms.
> >
> > Finally, the computational cost of the Neighborhood Search Extraction (NSE) method appears to be high and should be explicitly discussed in the paper.
> >
> > Given these concerns, I will maintain my current rating.

---

> ### Comment · Reviewer_JQ4Y · 2025-08-05
>
> Thanks for the further clarification and I appreciate authors' effort in rebuttal.
>
> I acknowledge that this work has potential and introduces some novel ideas. However, the current presentation significantly hinders understanding of its core contributions.
>
> **First, the contributions are not clearly articulated.**
> The paper builds on diffusion models (DMs), whose advantages over autoregressive models (ARMs) have been recognized in prior work (e.g., lines 48–54). However, it remains unclear what unique challenges arise when applying DMs to linguistic steganography, and how this work addresses them. The current narrative reads as: DMs are better and popular → we apply them to steganography → we achieve secure and robust performance. This gives the impression that the work is more of an application of known tools—a possible low-hanging fruit—rather than addressing a fundamental challenge.
>
> The authors could strengthen the framing by clarifying: DMs have clear advantages over ARMs → however, directly applying DMs to steganography introduces challenges, such as XXX (e.g., cumulative decoding errors) → we propose methods like robust position selection to overcome these challenges, and here are why we design like these and why they are effective. Furthermore, compared to robust position selection, the use of ECC feels more like a performance-enhancing add-on. It would help to explicitly distinguish between core contributions and auxiliary techniques. Otherwise, as noted by reviewers jzKv and 7ApW, readers may question the choice and necessity of specific components. My understanding is that people read papers to gain insigts more than better results, so identifying key challengs and show your motivations and design choices with sufficent experiments are important.
>
> **Second,  figures are powerful tools for conveying motivation and methodology.  However, in their current form, the figures fall short of supporting the main text.**
> For example, Figure 1 does not fully reflect the point made in the main text: `...impacts all messages of subsequent tokens due to the change in the conditional distribution.` The current version only shows distribution changes but omits the crucial consequence—message decoding errors. For readers unfamiliar with steganography, this obscures the practical impact of the propagation error. Including both the encoded and decoded message bits would make the figure more aligned with the claim and enhance clarity. Even in the recent clarification for Figure 2, how does the figure reflect `in the embedding phase, we proposed the concept of "Robust Position Selection" for DM. We only embed secret information at these positions to avoid cumulative errors, utilizing its ability to denoise different position in parallel and independently. In each step, we select a fixed embedding capacity according to the minimum entropy of the distributions, and use repetition codes as error correction codes (ECC) to correct errors while minimizing interference between different embedding positions.`
>
> **Third, I share other reviewers’ concerns about the limited experimental analysis.**
> It is important to disentangle where the performance gains come from—whether primarily from DMs, robust position selection, ECC, or neighborhood search—and what trade-offs each component entails. While the rebuttal begins to address these, a more thorough discussion supported by experiments would strengthen the paper.
>
> **Ultimately, readers turn to papers for insights, not just better results.**
> It’s important not only to report improved performance but to identify the key challenges, motivate the proposed solution, and clearly justify the design choices. This paper would benefit from a stronger emphasis on why certain methods were chosen and what insights they provide about the nature of steganography with diffusion models. **Overall, I believe this paper could have a stronger impact with revisions aimed at improving clarity and sufficient experiments and discussion.**

---

> > ### Author Response · Authors · 2025-08-06
> > **Reply to Reviewer JQ4Y #2**
> >
> > (continuing from the previous comment)
> > ...... We would like to modify the figures as follows:
> >
> > - In Figure 1, we ignored the fact that readers may not be able to associate the change in distribution with the extracted errors. We will include both the encoded and decoded message in the figure.
> > - In Figure 2, the left half is the steganographic embedding process, the right half is the steganographic extraction process, both of which are based on the DM generation process from top to bottom. At t=1, all positions' tokens are masked, while at t=0, all positions' tokens are denoised (unmasked), forming a complete sentence. We will supplement relevant information about RPS and ECC in the yellow and blue boxes in the figure.
> > - We plan to add a new figure to more clearly demonstrate how RPS and ECC are coupled and added into the DM generation process, which is an enlargement of the content within the yellow and blue boxes in Figure 2. This figure will provide a more detailed comparison of the differences in content embedded at robust positions and €€other non-robust positions. The figure will be placed in Section 4.2 as a visual representation of the content of formula 12.
> >
> > **Third, we are very willing to add all additional experiments of rebuttals to the revised paper.** Specifically, it includes robustness experiments of existing non-robust PSS methods after redundant encoding, robustness experiments of existing PSS methods directly transferred to DM, and comprehensive ablation experiments of all the components proposed. A more thorough discussion among those experiments would be added to the paper. Thank all the reviewers for the valuable questions; these experimental results will greatly strengthen the paper.
> >
> > In summary, we are very grateful for the valuable suggestions made by the reviewers, which have been very helpful in enhancing the clarity of our submission. We plan to make the following revision to the paper:
> >
> > 1. (Section Introduction): We will reiterate the parallel sampling characteristics of the DM model and its difference with ARM, and explain the advantages it brings to correct errors. We will supplement and emphasize the challenges encountered when directly introducing steganography into the DM model, as well as how we have utilized the advantages and addressed the challenges through the design of RPS and ECC. (See the aforementioned (introduction) and (Core contributions and auxiliary techniques)); we will modify Figure 1 to supplement the message bits of encoding and decoding. We will unify the expression of "Robust Position Embedding with Error Correction Coding" throughout the text, strengthening the connection of related concepts.
> > 2. (Section Methodology): We will reorganize Section 4, using subsection titles and bolded paragraph names to present concepts such as denoising positions, robust positions, redundant codes, and neighborhood search extraction in a clearer manner. For example:
> >     - We will modify the title of subsection 4.2 to "Robust Position Embedding with Error Correction Coding", to demonstrate the coupling of RPS and ECC.
> >     - We will directly define **denoised positions (lines 194-196)** as the subset of positions that are denoised (unmasked) in parallel at each reverse-diffusion step, while **robust positions** (line 224-232) as a subset of denoised positions that are capable of carrying enough bits to support repetition codes. And we will also provide a precise representation combining with formulas.
> >     - We will add a more detailed introduction about error localization and the strategies for determining the size of the neighborhood search window in the appendix. A hyperlink will be added to Section 4.2.
> > 3. (Section Experiments): The additional experiments in rebuttals will to be included to the revised paper, with some due to page limitation being placed in the appendix. The specific experiments include:
> >     - A complete ablation experiment for each proposed component;
> >     - An experiment of failure of migrating the existing methods to DM;
> >     - An experiment of failure of adding redundant encoding to the existing methods;
> >     - An ablation experiment on why to use repetition codes;
> >     - An experiment on robustness under more realistic attack scenarios, including mixed attacks and semantic attacks;
> >     - Security generalization experiments with different datasets;
> >     - An experiment on the computational time consumption of neighborhood search strategies;
> >     - An experiment on the quantitative estimation of provable robustness boundaries.

---

> > > ### Comment · Reviewer_JQ4Y · 2025-08-06
> > >
> > > Thanks for your response. Given the new clarification and the plan for revision, I'd like to raise my rating to a 4.
> > >
> > > I have a quick question regarding lines 63-65 of the text, where you mention, `we propose a concept of 'Robust Position Selection' for a DM, leveraging its ability to independently denoise tokens at different positions in parallel. We embed secret messages only at these positions to avoid cumulative errors.` However, in your recent rebuttal, you state that `the advantages brought by the parallel sampling feature of DM act on correcting errors rather than avoiding errors.` This appears to be a contradiction. Could you please clarify this further?

---

> > > > ### Author Response · Authors · 2025-08-07
> > > >
> > > > We apologize for the confusion, but this is a well-explained issue.
> > > >
> > > > - We mentioned in the text that we only embed in robust positions `to avoid cumulative errors`, where the **cumulative errors** correspond to the error propagation in Figure 1. Avoiding cumulative errors means that token errors that occur in a batch of robust positions will not affect the probability distribution of other tokens in the same batch predicted by the diffusion model.
> > > > - As mentioned in the rebuttal, `correcting errors rather than avoiding errors`, indicating that the selection of robust positions cannot prevent adversaries or the environment from tampering with the tokens in these positions. Due to the parallel denoising, the error propagation between robust positions is prevented, making the application of ECC possible and to correct the errors.
> > > >
> > > > It can be seen that the essence of these two different expressions is the same. We will **change the description of cumulative errors** to be consistent with the error propagation mentioned above to avoid ambiguity. At the same time, we will again **clarify the definition of robust positions** in the introduction to **avoid giving readers the wrong impression** that errors would not occur at robust positions.

---

> > > > ### Author Response · Authors · 2025-08-07
> > > >
> > > > Thank you again for acknowledging the new explanation and revision plan for the paper. Your patient guidance and valuable suggestions have been of great benefit to us. You have fully fulfilled your duties as a reviewer, helping us improve our submission.

---

> ### Author Response · Authors · 2025-08-06
> **Reply to Reviewer JQ4Y #1**
>
> Thank you very much for your detailed and patient comments. We are delighted to receive your recognition of the potentially strong impact and novel ideas in the paper. We are very willing to revise the submission to improve clarity and add sufficient experiments and discussion. Below are the responses and clarifications to the specific points you raised.
>
> **First, we would like to clarify the contributions of the paper.** The core goal of this paper is to achieve robust and provably secure steganography (PSS). As Reviewer 7ApW mentioned, previous PSS methods are often extremely brittle, while robust steganography methods offer no formal security. STEAD exploits the parallel denoising property of discrete diffusion models, proving that both guarantees can coexist.
>
> - **(Introduction) The use of the diffusion model is not the purpose itself but the means to achieve the purpose in this paper.** Therefore, in the introduction section of original submission, we proceed with the following process: `The current PSS methods are all based on ARM implementation → The ARM generation characteristics lead to excessive vulnerability → The parallel generation method of the DMs is promising to solve this problem → We implement robust provably security steganography based on DM`. However, we did indeed neglect to introduce the challenges of directly bringing the DMs into PSS. Specifically, **the advantages brought by the parallel sampling feature of DM act on correcting errors rather than avoiding errors.** As we demonstrated in our rebuttal to Reviewer jzKv and 7ApW, **directly deploying PSS to DM does not bring the expected robustness**, and, due to the strong binding between the distributions and the token positions, it will be **more fragile** than the PSS method based on ARM when faced with possible **position offsets** (such as additions, deletions, or token ambiguity). Therefore, it is precisely our designed algorithms and strategies, such as robust position selection, embedded error correction coding, and neighborhood search extraction, which were never seen in previous PSS methods, that together make it possible to achieve robust PSS with DM.
> - **(Core contributions and auxiliary techniques)** It needs to be clarified that RPS and ECC are highly coupled, and neither of them is an auxiliary technique. We first need to explain how DMs parallelly denoise masks, and then explain how RPS and ECC utilize this process to achieve robustness. Firstly, denoising at certain positions in each time step for DM is a random process. In a step, there may be **ZERO to MULTIPLE** positions that change from the MASK token to semantically meaningful tokens (line 185-189, formula 7). Since the denoised token at the current time step is the condition of the next time step (3rd line of formula 7), error correction must be completed within the current time step. This means that if the number of synchronized denoising locations is less than 3, for robust steganography, DM does not show a significant difference from ARM. Because there is no way to implement error correction. The number 3 is the minimum requirement for error correction codes. In this paper, the robust positions is defined in a way that makes them capable of carrying sufficient error-correcting information (as shown in last comment). That is to say, without ECC, there would be no robust position. And if robust positions are not filtered, due to the inability to embed sufficient information, ECC cannot perform its error correction function as expected.
> - **(Choice and necessity of specific components)** We apologize for any confusion caused to the readers and are committed to fully presenting the reason for our choices and additional experiments in the revised manuscript. Specifically, it includes the above mentioned explanation about how robust positions are selected, and a detailed explanation of why the repetition code is used, and why advanced robust coding methods like BCH may not be as effective in the current scenario.
>
> **Second, figures and their corresponding explanations would be modified.** We strongly agree with the point you raised: figures are powerful tools for conveying motivation and methods. A difficulty in the current presentation is that readers are hard to be simultaneously familiar with the generation process of ARM and DM, and the embedding and extraction process of PSS. Simultaneously displaying the generation process of the model and the steganographic process of our method clearly in the figures is something our article very much needs but has not done well. We would like to modify the figures as follows:
>
> (to be continued)

---

### Official Review · Reviewer_QWsN · 2025-06-30

**Clarity:** 2
**Significance:** 3
**Originality:** 3
**Rating:** 4
**Confidence:** 2

**Summary:**

This paper introduces a novel method for linguistic steganography that leverages diffusion language models (DMs) to achieve robustness and provable security. The contributions are:

1. **Robust Steganographic Framework**: The authors present a robust and provably secure linguistic steganography framework, STEAD,  using discrete diffusion language models. Besides the robustness, STEAD can use the parallel generation capabilities of DMs to mitigate error propagation issues.

2. **Error Correction Strategies**: To further improve the robustness, the authors employ error correction codes (ECC) and neighborhood search strategies during extraction to correct errors and handle position offsets.

3.  **Theoretical and Experimental Validation**: The authors provide extensive theoretical proofs and experimental results to demonstrate the security and robustness of STEAD.

**Questions:**

1. Can you introduce how to determine the denoising positions more clearly?

2. Lines 304-307, the authors stated the KLD of STEAD because STEAD will not modify the distribution. However, the way of sampling may affect the real `distribution`.  The authors need to explain if the sampling of STEAD will affect this.

**Ethical Concerns:**

["NO or VERY MINOR ethics concerns only"]

**Final Justification:**

I have provided my final justification.

**Limitations:**

yes

**Quality:**

3

**Strengths And Weaknesses:**

**Strengths**
1. Compared to the existing works, this work has discarded the traditional auto-regressive decoding paradigm, bringing higher efficiency and less error propagation.
2. Using the Diffusion Language Model is quite novel in this field.
3. The proposed STEAD is theoretically sound in robustness and security.
4. Nice Steganalysis results, where most results are near the golden line 50\%.
5. Nice efficiency results.


**Weaknesses**

1. Maybe due to my background, this work is not easy to follow.  The core idea and methodology were not introduced clearly.
2. How to select embedding (denoising） positions is crucial; however, this work lacks enough discussion about this. I can find the relevant part in Lines 216-232, but it seems incomplete.
3.  Limited comparison to standard methods. The embedding capacity of the proposed STEAD is still far behind the non-robust methods (typically $2-3\times 10^3$ per  $10^3$ tokens).  The authors should discuss (and conduct experiments) to compare `non-robust methods`  + `redundancy encoding`  vs 'robust methods', where the former adds a lot of redundancy/error correction codes to improve the robustness.
4. The Robustness against xxx results are impressive.  However, the settings of baselines are not clear. Hence, I am not sure if the comparison is fair.
5. The attack methods are also limited. The authors should compare more attack types.

---

> ### Author Rebuttal · Authors · 2025-07-30
>
> Thank you for your valuable comments. Here are our responses.
>
> 1. **Weakness 1: Maybe due to my background, this work is not easy to follow. The core idea and methodology were not introduced clearly.**
>
>     Response:
>
>     We would like to clarify that our method is composed of the following parts:
>
>     - Message-Driven Pseudo-Random Sampling: Secret bits are conveyed by adding a small, message-dependent offset to the pseudo-random number that drives sampling within each token distribution. Because the underlying distribution is unaltered, the stegotext remains provably indistinguishable from ordinary output.
>     - Robust Position Selection: At each reverse-diffusion step, STEAD identifies the subset of positions that are unmasked in parallel. These positions are independent of one another, making them suitable for isolated, error-tolerant embedding. When they are capable to carry enough bits, they are selected as robust positions.
>     - Error Correction Coding: Repetition codes are used to repetitively embed message across robust positions of a step, enabling correction of up to half the substitutions in that batch. Positions without payload serve as pseudo-random checks that can self-correct single-token substitutions.
>     - Neighborhood Search Extraction: A µ-neighborhood search strategy realigns the decoding process when global offsets arise from insertions or deletions.
>
>     In summary, we proposed a  stegosystem based on diffusion language models, which is simultaneously provably secure and robust. We will integrate this above explanation into Section 4 to explicitly illustrate the steganographic embedding and extraction process on DMs, ensuring full clarity.
>
> 2. **Weakness 2 & Question 1: How to select embedding (denoising） positions is crucial; however, this work lacks enough discussion about this. I can find the relevant part in Lines 216-232, but it seems incomplete.**
>
>     **Response:**
>
>     We would like to clarify the definition of the robust positions (for embedding) and denoising position.
>
>     - **Denoised positions (lines 194-196):** At each reverse-diffusion step, STEAD identifies the subset of positions that are denoised (unmasked) in parallel. These positions are selected independently via conditional probabilities (formula 7) during each step (consistent across cover and stego generation).
>     - **Robust positions (lines 224-232):** Robust positions are a subset of denoised positions that meet two criteria (formula 12):
>         1. The number of such positions in each step is at least 3 (to support repetition codes).
>         2. The minimum entropy of their distribution allows embedding at least 1 bit per position.
>
>         When the set of denoised positions at a certain step meets the above conditions, STEAD embeds the encoded message bits at these positions, making them robust embedding locations. We will modify the original text to make the above explanation clearer.
>
> 3. **Weakness 3: Limited comparison to standard methods. The embedding capacity of the proposed STEAD is still far behind the non-robust methods. The authors should discuss (and conduct experiments) to compare `non-robust methods` + `redundancy encoding` vs 'robust methods', where the former adds a lot of redundancy/error correction codes to improve the robustness.**
>
>     **Response:**
>
>     We acknowledge that STEAD’s embedding capacity lags behind non-robust methods. This aligns with the inherent trade-off in robust steganography: under provable security constraints (bound by information entropy), robust methods cannot exceed the capacity of non-robust counterparts.
>
>     Regarding "non-robust methods + redundancy coding" vs. "robust methods," we note a critical limitation: redundancy coding fails to make standard provably secure non-robust methods robust. Generative models (especially autoregressive ones) suffer from error propagation—tampering with one token disrupts the probability distribution of all subsequent sampling steps, as illustrated in Figure 1.
>
>     Experimental results with Discop (a standard non-robust method) + (20,1) repetition codes vs. STEAD confirm this:
>
>     | Method | Capacity (bit) | Correct (%) | Wrong (%) | Loss (%) | Decoded length (bit) |
>     | --- | --- | --- | --- | --- | --- |
>     | Discop + (20,1) Code | 59.70 | 1.14 | 0.89 | 97.97 | 1.21 |
>     | STEAD | 23.93 | 75.14 | 8.68 | 16.18 | **20.06** |
>
>     Using other error-correcting codes (e.g. BCH, LDPC) will not make this idea of directly combining non-robust methods with redundancy encoding feasible. We will add the above discussion to the revised paper.
>
> 4. **Weakness 4: The Robustness against xxx results are impressive. However, the settings of baselines are not clear. Hence, I am not sure if the comparison is fair.**
>
>     **Response:**
>
>     We appreciate the concern about the fairness of baseline comparisons.
>
>     To ensure consistency, we uniformly adopted advanced ARM models with identical parameter scales: Qwen2.5-7B and Deepseek-7B-base. For the generation process, **prompts and sampling settings** (temperature=1.0, top-p=0.95 for main experiments) were strictly aligned across all compared methods, including our STEAD. Additionally, the **attack settings (types and intensity)** in robustness experiments were standardized for all methods. The embedding capacity, embedding extraction time, and robustness results reported in our paper are all based on the average results of multiple repeated experiments in the IMDB dataset. We used the same fixed starting random seed for each comparison method, then updated the seed for each prompt and conducted multiple experiments in generation. Detailed information on these settings will be supplemented in Appendix D.
>
> 5. **Weakness 5: The attack methods are also limited. The authors should compare more attack types.**
>
>     **Response:**
>
>     To address this, we have extended the robustness evaluation to include two more challenging scenarios:
>
>     1. **Hybrid attacks** (combining token-level substitution, addition, and deletion) with both weak and strong intensity;
>     2. **Semantic-aware attacks** (synonym replacement) with perturbation rates of 0.01 and 0.05.
>
>     New results confirm that STEAD maintains superior robustness under these realistic scenarios, significantly outperforming other provably secure methods (discop+qw, ars+qw) across all metrics:
>
>     | Weak Mixed Attack | discop+qw | ars+qw | stead+diff |
>     | --- | --- | --- | --- |
>     | Correct | 16.92 | 34.99 | 97.59 |
>     | Wrong | 0.34 | 13.63 | 0.83 |
>     | Lost | 82.73 | 51.38 | 1.58 |
>
>     | Strong Mixed Attack | discop+qw | ars+qw | stead+diff |
>     | --- | --- | --- | --- |
>     | Correct | 2.57 | 12.65 | 75.14 |
>     | Wrong | 0.34 | 14.43 | 8.68 |
>     | Lost | 97.09 | 72.93 | 16.18 |
>
>     | Semantic-aware (0.01) | discop+qw | ars+qw | stead+diff |
>     | --- | --- | --- | --- |
>     | Correct | 43.60 | 50.44 | 99.43 |
>     | Wrong | 0.42 | 9.97 | 0.40 |
>     | Lost | 55.98 | 39.59 | 0.17 |
>
>     | Semantic-aware (0.05) | discop+qw | ars+qw | stead+diff |
>     | --- | --- | --- | --- |
>     | Correct | 10.27 | 22.60 | 89.16 |
>     | Wrong | 0.59 | 15.54 | 4.36 |
>     | Lost | 89.13 | 61.86 | 6.48 |
>
>     We will add a dedicated subsection "Real-World Attack Scenarios" in the Experiments section to detail these evaluations, further strengthening the validity of our experiments.
>
> 6. **Question 2: Lines 304-307, the authors stated the KLD of STEAD because STEAD will not modify the distribution. However, the way of sampling may affect the real `distribution`. The authors need to explain if the sampling of STEAD will affect this.**
>
>     Response:
>
>     While STEAD’s embedding algorithm does adjust the sampling process (as is standard in this field), it is designed to preserve the original token distribution.
>
>     In provably secure linguistic steganography, "covertext" refers to text generated via standard pseudo-random sampling from the language model, and "stegotext" to text generated through the steganographic embedding process. Security requires that the token distribution of stegotext (selected based on the hidden message) matches that of covertext (selected via random sampling).
>
>     As detailed in lines 255–281, we **formally prove** that for any polynomial-time distinguisher, the stego tokens chosen via STEAD’s message-driven sampling are **computationally indistinguishable** from cover tokens chosen via random sampling. This ensures that STEAD’s sampling method does not alter the underlying token distribution, maintaining consistency with random sampling.
>
>     Beyond the theoretical proof of computational indistinguishability and 0 KLD (preserving linguistic quality via PPL distribution), we supplemented steganalysis experiments. Trained detectors (FCN, R-BiLSTM-C, LSTMATT) show near-random accuracy (~50%) on distinguishing STEAD stegotexts from covertexts, validating security. We tested on a new dataset (C4) with 2000 cover-stego pairs. Results confirm consistent performance, with steganalyzers achieving accuracy close to random guessing:
>
>     | C4 dataset | FCN | R-BiLSTM-C | LSTMATT |
>     | --- | --- | --- | --- |
>     | accuracy | 49.92±2.79 | 49.08±2.66 | 50.25±0.54 |

---

> > ### Comment · Reviewer_QWsN · 2025-08-07
> > **Keep Rating Unchanged**
> >
> > Thanks for your response.
> >
> > This response has partially addressed my concerns and answered my questions.
> >
> > By checking the reviews from other reviewers,  it seems like all reviewers have the same rating (4). Thus, I think there is a major divergence among us, and I keep my recommendation.
> >
> > Finally, I have to say my confidence is only 2 again.

---

> > > ### Comment · Reviewer_QWsN · 2025-08-07
> > > **Reasons**
> > >
> > > I gave a rating of 4 because I can not make a very clear judgment on many points.
> > >
> > > If I ignore this, maybe I will give a rating of 3.

---

> > > ### Author Response · Authors · 2025-08-07
> > >
> > > We are grateful for your review and for your feedback. We are glad that this response has addressed your concerns and questions.
> > >
> > > At the same time, we also noticed that all the reviewers have given a 4-rating evaluation — the consistency of this feedback has helped us more clearly understand the current positioning and direction for improvement of the submission. We fully respect your decision to maintain the recommendation.
> > >
> > > Once again, thank you for your patient guidance and professional support; your feedback is an important driving force for us to improve the quality of the paper.

---

### Official Review · Reviewer_7ApW · 2025-07-01

**Clarity:** 4
**Significance:** 3
**Originality:** 3
**Rating:** 4
**Confidence:** 3

**Summary:**

The paper targets a long-standing challenge in linguistic steganography: embedding secret messages in generated text so that
1. the stegotext is computationally indistinguishable from ordinary model output (provable security) and
2. the hidden message can still be recovered even if an adversary inserts, deletes, or substitutes tokens (robustness).
Existing provably secure techniques rely on autoregressive language models (ARMs). Because ARMs generate tokens strictly left-to-right, any tampering with one token disrupts the conditional distribution of all subsequent tokens, causing catastrophic error propagation during decoding. As a result, prior ARM-based systems are fragile under active attacks.
To overcome this, the authors leverage the different generation paradigm of discrete diffusion language models (DMs), which denoise many token positions in parallel. They introduce STEAD, a stegosystem that embeds information only in those sets of tokens that are denoised simultaneously, thereby avoiding the cascading-error problem.
Key components and contributions:
1. Robust Position Selection
At each reverse-diffusion step, STEAD identifies the subset of positions that are unmasked and denoised in parallel. These positions are statistically independent of one another, making them suitable for isolated, error-tolerant embedding.
2. Message-Driven Pseudo-Random Sampling
Secret bits are conveyed by adding a small, message-dependent offset to the pseudo-random number that drives sampling within each token distribution. Because the underlying distribution is unaltered, the stegotext remains provably indistinguishable from ordinary DM output.
3. Built-in Error Correction
The same message fragment is embedded redundantly across all denoised positions of a step (a repetition code), enabling correction of up to half the substitutions in that batch.
4. Pseudo-Random and Neighborhood Corrections
Positions without payload serve as pseudo-random parity checks that can self-correct single-token substitutions. A µ-neighborhood search strategy realigns the decoding process when global offsets arise from insertions or deletions.
Collectively, the work demonstrates that diffusion language models enable a practical stegosystem that is simultaneously provably secure and markedly more robust to active text tampering than prior autoregressive approaches.

**Questions:**

1. Have you ever calculated the empirical distribution of Ns/L on test corpus?  In addition, how does the message-recovery rate behave in instances where the condition 2(α+β+γ)<Ns/L2 is not satisfied?
2. Have you evaluated the proposed method on diffusion language models with additional parameter scales or alternative architectures, and have you repeated the experiments under multiple random seeds to demonstrate the statistical robustness of your results?
3. What motivated your choice of a repetition code instead of a BCH code in the implementation? This decision seems to impose a capacity penalty.

**Ethical Concerns:**

["NO or VERY MINOR ethics concerns only"]

**Final Justification:**

4: Borderline accept

**Limitations:**

yes

**Paper Formatting Concerns:**

No formatting issues in this paper

**Quality:**

3

**Strengths And Weaknesses:**

Strengths
1. Fresh Perspective: Diffusion + Provable Security + Robustness in a Single Framework
 - Earlier work had to pick two of the three:
   -   ARM-based methods → provably secure but extremely brittle.
   -   Robust-LLM methods → sturdy but offer no formal security.
 - STEAD exploits the parallel denoising property of discrete diffusion models, proving that both guarantees can coexist. This insight is likely to inspire follow-ups in steganography and in other security uses of diffusion models (watermarking, fingerprinting, etc.).
2. Large and Intuitive Empirical Gains
Outperforms best ARM baseline by large margins on capacity and robustness while leaving perceptual quality and detectability unchanged.
  - Capacity: 84 bit / 1 k tokens vs. 14 bit for the strongest ARM baseline (≈ 6×).
  - Robustness: 10 % random substitutions → 99 % full recovery, while three ARM baselines collapse; same story for insertions/deletions.
 Provides both formal guarantees and empirical evidence that the method is secure and resilient to common active attacks.
3.  Low Integration Cost
 - Requires no model retraining—just swap the sampling function; applicable to any discrete diffusion LM.
 - Core code amounts to two concise algorithms (embed/extract), which makes community reproduction and extension easy.

Weaknesses
1. Generality of Experiments Is Unclear
 -   All results are obtained with a single diffusion model (Dream-7B).  It remains unclear whether the claimed advantages persist across architectures scales, and masking schedules.
-   No repeated runs or error bars—stability is unknown.
2. Coarse Capacity–Efficiency Trade-off
      The system relies on naive repetition coding, which, while analytically convenient, is bandwidth-inefficient; sophisticated short block codes (BCH, LDPC, Polar) could yield substantially better rates.  The computational footprint of decoding with a wide µ-neighbourhood search and of multi-step diffusion sampling is not quantified (latency, GPU-hours), casting doubt on practical deployability.
3. Some Theoretical Assumptions Lack Empirical Support
   The robustness bound depends on the inequality 2(α+β+γ)<Ns/L.  The manuscript does not provide statistics on Ns  across prompts, leaving open whether the condition routinely holds.
4. Potentially Incomplete Baseline Comparison
 All baselines are autoregressive.  Although Appendix A notes that Discop ported to diffusion fails catastrophically, no quantitative results or analysis are provided.  A head-to-head comparison with contemporary robust-but-insecure diffusion methods (e.g., Wu et al.) on a combined robustness and detectability axis is missing, which may raise fairness concerns.

---

> ### Author Rebuttal · Authors · 2025-07-30
>
> Thank you for your detailed comments. We strongly agree with your view that **"this insight is likely to inspire follow-ups in steganography and in other security uses of diffusion models (watermarking, fingerprinting, etc.)."** Our responses are list below:
>
> 1. **Weakness 1 & Question 2:** Generality of Experiments Is Unclear.
>
>     **W1.1 All results are obtained with a single diffusion model (Dream-7B). It remains unclear whether the claimed advantages persist across architectures scales, and masking schedules.**
>
>     **Response:**
>
>     - **Regarding architectures scales:** The use of Dream-7B stems from the current landscape of diffusion language models—large-scale pre-trained alternatives remain limited, and Dream-7B is a state-of-the-art (SOTA) model with publicly available 7B-scale weights.
>
>         Our steganographic method operates by modifying only the model’s **sampling process**, making it largely independent of parameter scale. Thus, the core advantages (robustness and security) are not tied to specific model sizes. If larger diffusion models are released, we will extend our research to analyze how model type and weight size impact embedding time and capacity.
>
>     - **Regarding masking schedules:** Our method retains the diffusion model’s native denoising strategy (random denoising based on specific probabilities). This design ensures compatibility across different diffusion architectures and aligns with the security requirements of provably secure steganography.
>
>     **W1.2  No repeated runs or error bars—stability is unknown.**
>
>     **Response:**
>
>     We acknowledge the concern about result stability. All reported metrics—embedding capacity, extraction time, and robustness—**are averaged over multiple repeated experiments** on the IMDB dataset.
>     To ensure consistency, each comparison method used a fixed initial random seed, with seeds updated per prompt across repeated runs. Text generation for each prompt and the random attacks applied were fully independent. The results in the paper reflect statistical averages of these independent experiments.
>     we will explicitly state this experimental design in the revised version and add error bars to clarify result stability, enhancing reproducibility.
>
> 2. **Weakness 2 & Question 3: Coarse Capacity–Efficiency Trade-off**
>
>     **Weakness 2.1 The system relies on naive repetition coding, which, while analytically convenient, is bandwidth-inefficient; sophisticated short block codes (BCH, LDPC, Polar) could yield substantially better rates.**
>
>     **Response:**
>
>     As the first work implementing provably secure steganography on DM architectures, our choice of repetition codes is driven by the unique challenges of robust text steganography rather than analytical convenience.
>
>     In robust provably secure steganography, text token tampering differs from traditional bit flipping: each token may carry multiple bits, and tampering can cause random errors across all bits in the token. Moreover, due to the dependency on predicted distributions in provably secure designs, incomplete token restoration disrupts subsequent sampling distributions, invalidating message extraction.
>
>     The scenario here involves **short code lengths with high error-correction demands**, as shown in our supplementary data:
>
>     | Dataset | IMDB | C4 |
>     | --- | --- | --- |
>     | Code Length | 6.28 | 6.69 |
>     | Error Length | 1.92 | 2.05 |
>
>     Assuming at most one token error per generation step (a low-intensity attack), the required error-correcting bits relative to code length make linear codes (BCH, LDPC, Polar) degenerate to repetition codes under (n,1) parameters. This prioritizes robustness, which is critical for maintaining distribution consistency in sampling.
>
>     We will clarify this reasoning in the paper and note that future work will explore efficient encoding methods tailored to robust text steganography, balancing capacity and error correction.
>
>     **Weakness 2.2 The computational footprint of decoding with a wide µ-neighbourhood search and of multi-step diffusion sampling is not quantified (latency, GPU-hours), casting doubt on practical deployability.**
>
>     **Response:**
>
>     We appreciate the focus on computational efficiency for practical deployment. Regarding the decoding process with µ-neighbourhood search, we tested the time consumption and accuracy under different µ sizes under a strong mixed attack (token replacement, addition, deletion with intensity 0.1,+3,-3), as shown in the table below:
>
>     | **Strong Mixed Attack（0.1,+3,-3）** | mu=1 | mu=2 | mu=4 |
>     | --- | --- | --- | --- |
>     | total time (s) | 48.22 | 59.46 | 62.12 |
>     | decode time (s/bit) | 2.80 | 2.78 | 2.77 |
>     | Correct | 56.85 | 82.58 | 90.58 |
>     | Wrong | 15.05 | 6.19 | 2.77 |
>     | Lost | 28.10 | 11.23 | 6.65 |
>
>     Notably, while larger µ improves accuracy, the time per effective bit remains stable (even slightly decreases).
>
>     For multi-step diffusion sampling, we measured the computational footprint using a single NVIDIA 4090 GPU. The average sampling time per 512-token stegotext is 38.22 seconds. The average time cost of embedding algorithm is 1.93 seconds. The results show that most of the time consumed in generating stego comes from the multi-step inference process of the model itself, making actual deployment feasible.
>
> 3. **Weakness 3 & Question 1: Some Theoretical Assumptions Lack Empirical Support The robustness bound depends on the inequality 2(α+β+γ)<Ns/L. The manuscript does not provide statistics on Ns across prompts, leaving open whether the condition routinely holds.**
>
>     **Response:**
>
>     We tested the average number of occurrences of different values of Ns in a 512-length token sequence generation under various prompts (using different IMDB datasets and C4 datasets). Due to the repetition code we adopt, embedding is required only when N is greater than 3. It can be seen that although the distribution of Ns is different, the minimum value of Ns is always 3.
>
>     | Dataset | Ns=3 | Ns=4 | Ns=5 | Ns=6 |
>     | --- | --- | --- | --- | --- |
>     | IMDB | 16.2 | 3.3 | 0.6 | 0.1 |
>     | C4 | 13.2 | 2.8 | 0.45 | 0.15 |
>
>     The theoretical assumption 2(α + β + γ) < mins(Ns/L)) defines the boundary for **perfect 100% extraction accuracy**. In practice, this boundary is strict and rarely met in complex real-world scenarios. For example, in generating 512-length text, minN*s* is typically 3, leading to a theoretical tampering threshold of ~0.3% (i.e., 1.5 tokens out of 512). This strict boundary is difficult to satisfy under realistic attacks. However, this does not invalidate message extraction beyond the boundary. Experimental results (Figures 5, 7, 8) show a **gradual rather than abrupt decline** in accuracy:
>
>     - At low tampering intensities (below the boundary), extraction accuracy remains nearly 100%.
>     - As tampering exceeds the boundary, accuracy decreases steadily but retains practical utility (e.g., 74.69% correct extraction even with 10 insertions in 500 samples, as shown in prior robustness tests).
>
>     This aligns with real-world language environments, where attacks vary in intensity but rarely cause extreme tampering. The theoretical bound serves as a benchmark for optimal performance, while empirical results demonstrate the method’s resilience beyond this idealized scenario. We will clarify this distinction between theoretical guarantees and practical performance in the revised manuscript.
>
> 4. **Weakness 4: Potentially Incomplete Baseline Comparison All baselines are autoregressive. A head-to-head comparison with contemporary robust-but-insecure diffusion methods (e.g., Wu et al.) on a combined robustness and detectability axis is missing.**
>
>     **Response:**
>
>     We recognize the importance of comprehensive baseline comparisons. As **the first diffusion model (DM)-based linguistic steganography method with both provable security and robustness**, STEAD currently lacks direct peers—no other DM-based methods with these dual properties exist for comparison.
>
>     To address this, we supplemented experiments by adapting Discop (a representative autoregressive model (ARM)-based provably secure method) to the DM framework (see Appendix E.1). Results confirm that naive migration of ARM methods to DMs fails to achieve robustness, as shown in the table below:
>
>     |  | Discop+DM | STEAD |
>     | --- | --- | --- |
>     | Correct | 0.82 | 98.89 |
>     | Wrong | 0.76 | 0.50 |
>     | Lost | 98.42 | 0.61 |
>     | Success rate | 0.0 | 99.49 |
>
>     Regarding GTSD (Wu et al., submitted to arXiv on 28 Apr 2025), it is our **contemporaneous work**. To ensure the sufficiency of the current situation described in this paper, we added relevant references, and the paper was officially published after our submission (24 June 2025), so we did not conduct a comparison. Notably, GTSD and our method (STEAD) differ fundamentally:
>
>     - GTSD is **empirically secure**. Its diffusion model requires dataset-specific training, and the embedding process introduces non-zero KL divergence, leading to detectable distribution differences (as reported, steganalysis detectors achieve ~10% accuracy above random guessing). This leaves it vulnerable to evolving steganalysis techniques.
>
>     - In contrast, STEAD is **provably secure**, theoretically guaranteeing indistinguishability between stego and cover data, eliminating detection risks—a key focus in current steganography research.
>
>     They are two types of methods: **empirically secure** and **theoretically secure**. So a direct quantitative comparison is not appropriate. We will add the above qualitative analysis to our revised paper.

---

> > ### Comment · Reviewer_7ApW · 2025-08-06
> >
> > Thank you for your rebuttal and clarification. The authors' response has adequately addressed my concerns. The authors have also promised to include error bars in the revised paper. While Dream-7B is indeed a state-of-the-art model, I believe that conducting experiments on models with different architectures and varying scales would further validate the generalization and robustness of the proposed method, aligning it more closely with the theoretical justifications.
> >
> > Additionally, while their explanation of GTSD's empirical limitations is valid, incorporating partial quantitative comparisons would further strengthen their argument.
> >
> > Overall, I maintain my positive rating as borderline accept.

---

> > > ### Author Response · Authors · 2025-08-07
> > >
> > > Thank you very much for your valuable suggestions and recognition of the paper. We are glad to address your concerns. We will also include the deployment of diffusion models with different architectures and scales, as well as a quantitative comparison of GTSD, in the plan for revising the paper.

---

### Official Review · Reviewer_jzKv · 2025-07-03

**Clarity:** 2
**Significance:** 3
**Originality:** 3
**Rating:** 4
**Confidence:** 4

**Summary:**

This paper proposes a language steganography framework named STEAD, aiming to address the poor robustness of existing PSLS methods based on ARMs when facing active attacks (such as substitution, insertion, and deletion). STEAD employs a DM as its generator, independently embedding information at multiple positions at specific time steps. It combines repetition codes as simple ECC, and introduces a neighborhood search strategy to handle token position shifts caused by insertions and deletions. The paper experimentally compares it with existing ARM-based methods, claiming advantages in terms of capacity, security, and robustness.

**Questions:**

In Weaknesses.

**Ethical Concerns:**

["NO or VERY MINOR ethics concerns only"]

**Final Justification:**

The authors addressed my concerns through additional experiments in their response, including the addition of baseline comparisons and ablation studies, expanded robustness evaluations across more scenarios, a larger-scale experimental setup, and the inclusion of new datasets. I appreciate their efforts and the way they responded to the issues I raised. I have decided to maintain my original score.

**Limitations:**

Yes.

**Quality:**

3

**Strengths And Weaknesses:**

Strengths:

The core contribution of this paper lies in identifying and leveraging the inherent parallelism advantage of DMs over ARMs in their generation mechanism, thereby providing a theoretical foundation for building robust steganography systems capable of resisting error propagation. Building upon this, the authors designed a systematic solution comprising "robust position selection," repetition error correction codes, and neighborhood search decoding, aimed at simultaneously addressing multiple attack types such as substitution, insertion, and deletion. Experimental results demonstrate the advantages of the proposed method over existing approaches.

Weaknesses:

The depth and breadth of the paper's experimental validation are insufficient. Firstly, the baseline comparisons in the experiments are unfair. Comparing the new DM-based method with older ARM-based methods in terms of robustness largely reflects the inherent differences between the two generative paradigms (DM vs. ARM), rather than the superiority of the STEAD method itself within the DM framework. Comparisons should instead include other potential robust steganography baseline methods that are also based on DMs. Secondly, the attack models employed for robustness evaluation are overly simplistic, testing different attack types independently only under uniformly random attack patterns. There is a lack of performance evaluation in more challenging, semantically aware, or hybrid attack scenarios. Finally, the experimental scale (200 text samples) is relatively small, and the security analysis is not comprehensive enough, which limits the statistical significance and generalizability of the experimental conclusions.

The paper lacks sufficient justification for some key design choices. For instance, the paper uses the simplest repetition codes as the error correction mechanism but does not explain why more efficient algebraic codes (e.g., BCH codes) were not chosen, nor does it analyze the potential trade-offs that different error correction codes might introduce for embedding capacity and error correction performance. Similarly, for the key component on the decoding end—neighborhood search—the strategy for setting its window size μ is unclear. This parameter directly affects decoding complexity and accuracy, yet the paper does not discuss in detail its selection rationale or dynamic adjustment mechanism, leaving a core aspect of the method lacking transparency and reproducibility.

Are there gaps between the assumptions of the paper's theoretical proofs (e.g., 2(α + β + γ) < mins(Ns/L)) and real-world attack scenarios? To what extent do these theoretical premises hold true in complex real-world language environments?

---

> ### Author Rebuttal · Authors · 2025-07-30
>
> Thank you for your valuable comments. Here are our responses.
>
> 1. **Weakness 1:** The depth and breadth of the paper's experimental validation are insufficient.
>
>     **W1.1 Unfair baseline comparisons**
>
>     **Response:**
>
>     We acknowledge the concern about baseline fairness. However, **STEAD is the first DM-based linguistic steganography method with provable security and robustness**—to date, no other DM-based methods with these dual properties exist for direct comparison.
>
>     To address this, we supplemented comparisons by migrating a representative ARM-based provably secure method (Discop) to the DM framework (see Appendix E.1). Results show that simply adapting ARM methods to DMs fails to achieve robustness (success rate = 0.0% vs. STEAD’s 99.49% under token segmentation ambiguity, as shown below).
>
>     |  | Discop+DM | STEAD |
>     | --- | --- | --- |
>     | Correct | 0.82 | 98.89 |
>     | Wrong | 0.76 | 0.50 |
>     | Lost | 98.42 | 0.61 |
>     | succeed rate | 0.0 | 99.49 |
>
>     Ablation studies further validate that STEAD’s robustness stems from DM-specific innovations: robust position selection (RPS) + error-correcting codes (ECC), and neighborhood search extraction (NSE) collectively contribute to its superiority, rather than relying solely on the DM paradigm.
>
>     |  | Replacement 0.2 | Insertion 10 | Deletion 10 |
>     | --- | --- | --- | --- |
>     | w/o RPS+ECC, NSE | 31.03/30.27/38.70 | 7.73/7.67/84.60 | 7.67/7.72/84.61 |
>     | w/o NSE | 95.22/1.77/3.01 | 6.85/8.89/84.26 | 4.96/8.47/86.57 |
>     | STEAD | 94.87/1.07/4.06 | 78.31/8.76/12.92 | 88.11/6.10/5.79 |
>
>     **W1.2 Overly simplistic attack models**
>
>     **Response:**
>
>     We appreciate the insight regarding attack model complexity. To address this, we have extended the robustness evaluation to include two more challenging scenarios:
>
>     1. **Hybrid attacks** (combining token-level replacement, addition, and deletion) with both weak and strong intensity;
>     2. **Semantic-aware attacks** (synonym replacement) with perturbation rates of 0.01 and 0.05.
>
>     New results confirm that STEAD maintains superior robustness under these realistic scenarios, significantly outperforming other provably secure methods (discop+qw, ars+qw) across all metrics:
>
>     | Weak Mixed Attack | discop+qw | ars+qw | stead+diff |
>     | --- | --- | --- | --- |
>     | Correct | 16.92 | 34.99 | 97.59 |
>     | Wrong | 0.34 | 13.63 | 0.83 |
>     | Lost | 82.73 | 51.38 | 1.58 |
>
>     | Strong Mixed Attack | discop+qw | ars+qw | stead+diff |
>     | --- | --- | --- | --- |
>     | Correct | 2.57 | 12.65 | 75.14 |
>     | Wrong | 0.34 | 14.43 | 8.68 |
>     | Lost | 97.09 | 72.93 | 16.18 |
>
>     | Semantic-aware (0.01) | discop+qw | ars+qw | stead+diff |
>     | --- | --- | --- | --- |
>     | Correct | 43.60 | 50.44 | 99.43 |
>     | Wrong | 0.42 | 9.97 | 0.40 |
>     | Lost | 55.98 | 39.59 | 0.17 |
>
>     | Semantic-aware (0.05) | discop+qw | ars+qw | stead+diff |
>     | --- | --- | --- | --- |
>     | Correct | 10.27 | 22.60 | 89.16 |
>     | Wrong | 0.59 | 15.54 | 4.36 |
>     | Lost | 89.13 | 61.86 | 6.48 |
>
>     We will add a dedicated subsection "Real-World Attack Scenarios" in the Experiments section to detail these evaluations, further strengthening the validity of our findings.
>
>     **W1.3 Small experimental scale and incomplete security analysis**
>
>     **Response:**
>
>     We appreciate the feedback on experimental scale and security analysis. To address this, we have taken the following steps:
>
>     1. **Expanded experimental scale**: We increased the sample size for robustness testing under random insertion to 500 stegotexts. Results (shown below) are consistent with the original findings, confirming statistical stability:
>
>
>         | Insertion | 2 | 4 | 6 | 8 | 10 |
>         | --- | --- | --- | --- | --- | --- |
>         | Correct | 99.48 | 94.82 | 92.29 | 79.61 | 74.69 |
>         | Wrong | 0.15 | 2.42 | 3.10 | 6.92 | 9.23 |
>         | Lost | 0.36 | 2.76 | 4.61 | 13.47 | 16.08 |
>     2. **Enhanced security analysis**: Beyond the theoretical proof of computational indistinguishability and 0 KLD (preserving linguistic quality via PPL distribution), we supplemented steganalysis experiments. Trained detectors (FCN, R-BiLSTM-C, LSTMATT) show near-random accuracy (~50%) on distinguishing STEAD stegotexts from covertexts, validating security.
>     3. **Improved generalizability**: We tested on a new dataset (C4) with 2000 cover-stego pairs. Results confirm consistent performance, with steganalyzers achieving accuracy close to random guessing:
>
>         | C4 dataset | FCN | R-BiLSTM-C | LSTMATT |
>         | --- | --- | --- | --- |
>         | accuracy | 49.92±2.79 | 49.08±2.66 | 50.25±0.54 |
>
>     These extensions strengthen the statistical significance and generalizability of our conclusions.
>
> 2. **Weakness 2:** The paper lacks sufficient justification for some key design choices.
>
>     **W2.1 Use of repetition codes over algebraic codes**
>
>     **Response:**
>
>     The use of repetition codes in STEAD is motivated by the unique characteristics of robust provably secure text steganography:
>
>     1. **Scenario-specific requirements**: Unlike traditional bit-flipping in lossy channels, text token tampering can corrupt multiple bits simultaneously (since each token carries multiple bits). For provably secure steganography, full token recovery is critical—any residual error propagates to subsequent extraction steps. This demands a mechanism robust to complete token-level failures.
>     2. **Constraints on code parameters**: In STEAD’s one-step generation, robust positions support very short code lengths with high error-correction demands. Supplementary data (below) shows that under typical conditions (assuming ≤1 token error per step), linear codes degenerate to repetition codes when parameters approach (n, 1):
>
>
>         | Dataset | IMDB | C4 |
>         | --- | --- | --- |
>         | Code Length | 6.28 | 6.69 |
>         | Error Length | 1.92 | 2.05 |
>     3. **Trade-off clarification**: We acknowledge the trade-off between robustness and embedding capacity. Repetition codes were chosen to prioritize maximum robustness, which is critical for the first DM-based provably secure method. While algebraic codes (e.g., BCH) offer efficiency in longer codes, they underperform in our short-code, high-error scenario.
>
>     We will expand this analysis in the revised paper and note that exploring adaptive coding schemes for varied attack intensities is a key direction for future work.
>
>     **W2.2 Neighborhood search window size (μ)**
>
>     **Response:**
>
>     The window size μ is dynamically adjusted based on the tampered text length: μ = max(2, |L - L'|), where L denotes the original sequence length and L' is the length after tampering. This strategy balances adaptability to varying attack intensities (e.g., insertion/deletion) while avoiding excessive computational overhead.
>
>     To validate this setting, we tested decoding performance under different fixed μ values using the strong mixed attack scenario. Results (below) show that the larger μ achieves higher accuracy without significant time overhead per effective bit:
>
>     |  | mu=1 | mu=2 | mu=4 |
>     | --- | --- | --- | --- |
>     | decode time（s） | 48.22 | 59.46 | 62.12 |
>     | decode time（s/bit) | 2.80 | 2.78 | 2.77 |
>     | Correct | 56.85 | 82.58 | 90.58 |
>     | Wrong | 15.05 | 6.19 | 2.77 |
>     | Lost | 28.10 | 11.23 | 6.65 |
>
>     We will include this dynamic adjustment logic and parameter analysis in the revised manuscript, along with implementation details to ensure reproducibility.
>
> 3. **Weakness 3: Gaps between theoretical assumptions and real-world scenarios**
>
>     **Response:**
>
>     The theoretical assumption 2(α + β + γ) < mins(Ns/L)) defines the boundary for **perfect 100% extraction accuracy**. In practice, this boundary is strict and rarely met in complex real-world scenarios. For example, in generating 512-length text, minN*s* is typically 3, leading to a theoretical tampering threshold of ~0.3% (i.e., 1.5 tokens out of 512). This strict boundary is difficult to satisfy under realistic attacks. However, this does not invalidate message extraction beyond the boundary. Experimental results (Figures 5, 7, 8) show a **gradual rather than abrupt decline** in accuracy:
>
>     - At low tampering intensities (below the boundary), extraction accuracy remains nearly 100%.
>     - As tampering exceeds the boundary, accuracy decreases steadily but retains practical utility (e.g., 74.69% correct rate even with 10 insertions, as shown in prior robustness tests).
>
>     This aligns with real-world language environments, where attacks vary in intensity but rarely cause extreme tampering. The theoretical bound serves as a benchmark for optimal performance, while empirical results demonstrate the method’s resilience beyond this idealized scenario. We will clarify this distinction between theoretical guarantees and practical performance in the revised paper.

---

### Note · Authors · 2025-08-12

We sincerely thank the AC and reviewers for their thoughtful and constructive feedback.

In the original submission, the reviewers recognized several key strengths of our work:

1. **Novelty:** Based on the inherent parallelism advantage of diffusion language models, it provides a theoretical foundation for building robust secure steganography systems capable of resisting error propagation.
2. **Theoretical Guarantee for Security and Robustness**: It breaks through the limitation of "difficulty in achieving both security and robustness" in existing research, providing theoretical reliability.
3. **Excellent Experimental Results:** The steganalysis results are close to the 50% golden line, and can effectively resist common attacks such as substitution, insertion, and deletion, combining theoretical reliability and experimental effectiveness.

The main points raised for further evaluation were:

1. More discussion and comparison with one contemporaneous work, GTSD.
2. Assessment of the effectiveness of the proposed components.
3. Assessment of the computational efficiency of the robust enhancement strategy.
4. Robustness under a more realistic tampering scenarios.
5. Clarity of the methodology section.

These points **focus on enhancing the clarity of the paper, and expanding the validation of STEAD’s applicability, rather than questioning its core novelty, contributions, or effectiveness**. All have been thoroughly addressed during the rebuttal period, with additional experiments and clarifications provided. **Reviewers have acknowledged these improvements，and they all have consistently given the paper positive ratings.**

We believe the revised work delivers a **novel, impactful, and rigorously validated** solution to robust provably secure steganography.

---

### Decision · Program_Chairs · 2025-09-17

**Decision:**

Accept (poster)

**Comment:**

This paper presents STEAD, an approach for linguistic steganography that uses the parallel generation abilities of diffusion language models (DMs) to achieve both provable security and robustness against attacks. The core idea is to embed information in independently denoised "robust positions," using error-correction to withstand tampering. This approach aims to solve the error propagation issues that plague prior approaches done on auto-regressive based models.

The reviewers all recognized the idea of using DMs for this problem, and the work has potential to inspire further research. The authors were very committed during the rebuttal, engaging thoroughly with feedback and providing substantial new data and results to address reviewer concerns.

I do have some concerns about the number of changes and new experiments conducted during the rebuttal period suggests the work was not ready for peer review. The authors had to add fundamental components to their evaluation, including ablation studies to validate their own contributions, new baselines (porting an ARM method to a DM), tests against more realistic hybrid and semantic attacks, and security analysis on an entirely new (and much larger) dataset. That is a lot of changes, and I hope that all of these will make it into the next version of the paper. I would also strongly recommend a comparison with GTSD as pointed out in the reviews, even if it does not have all the same properties as STEAD, a comparison is still useful. I realize this is concurrent work, but would strengthen the paper and address the concern about how much of the gains are from using a DM model and how much are from STEAD.